# Deep analysis of CD4 T cells in the rhesus CNS during SIV infection

**Sonny R. Elizaldi[1], Anil Verma[2], Zhong-Min Ma[3], Sean Ott[3], Dhivyaa Rajasundaram[4], Chase E. Hawes[1], Yashavanth Shaan Lakshmanappa[3], Mackenzie L. Cottrell[5], Angela D. M. Kashuba[5], Zandrea Ambrose[6], Jeffrey D. Lifson[7], John H. Morrison[3,8], Smita S. Iyer** [2,3,9]*

**1** Graduate Group in Immunology, UC Davis, California, United States of America, **2** Department of Pathology, School of Medicine, University of Pittsburgh, Pennsylvania, United States of America, **3** California National Primate Research Center, UC Davis, California, United States of America, **4** Department of Pediatrics, School of Medicine, University of Pittsburgh, Pennsylvania, United States of America, **5** Eshelman School of Pharmacy, University of North Carolina, Chapel Hill, North Carolina, United States of America, **6** Department of Microbiology and Molecular Genetics, School of Medicine, University of Pittsburgh, Pennsylvania, United States of America, **7** AIDS and Cancer Virus Program, Frederick National Laboratory, Frederick, Maryland, United States of America, **8** Department of Neurology, School of Medicine, UC Davis, California, United States of America, **9** Department of Pathology, Microbiology, and Immunology, School of Veterinary Medicine, UC Davis, California, United States of America

* siyer.3@pitt.edu

**Data Availability Statement:** RNA-seq dataset is accessible at GSE221815. https://www.ncbi.nlm.nih.gov/geo/query/acc.cgi?acc=GSE221815.

**Funding:** This work was supported by the National Institutes of Health (K01OD023034, RF1AG06001,

## Abstract

Virologic suppression with antiretroviral therapy (ART) has significantly improved health outcomes for people living with HIV, yet challenges related to chronic inflammation in the central nervous system (CNS)—known as Neuro-HIV- persist. As primary targets for HIV-1 with the ability to survey and populate the CNS and interact with myeloid cells to co-ordinate neuroinflammation, CD4 T cells are pivotal in Neuro-HIV. Despite their importance, our understanding of CD4 T cell distribution in virus-targeted CNS tissues, their response to infection, and potential recovery following initiation of ART remain limited. To address these gaps, we studied ten SIVmac251-infected rhesus macaques using an ART regimen simulating suboptimal adherence. We evaluated four macaques during the acute phase pre-ART and six during the chronic phase. Our data revealed that HIV target CCR5+ CD4 T cells inhabit both the brain parenchyma and adjacent CNS tissues, encompassing choroid plexus stroma, dura mater, and the skull bone marrow. Aligning with the known susceptibility of CCR5+ CD4 T cells to viral infection and their presence within the CNS, high levels of viral RNA were detected in the brain parenchyma and its border tissues during acute SIV infection. Single-cell RNA sequencing of CD45+ cells from the brain revealed colocalization of viral transcripts within CD4 clusters and significant activation of antiviral molecules and specific effector programs within T cells, indicating CNS CD4 T cell engagement during infection. Acute infection led to marked imbalance in the CNS CD4/CD8 ratio which persisted into the chronic phase. These observations underscore the functional involvement of CD4 T cells within the CNS during SIV infection, enhancing our understanding of their role in establishing CNS viral presence. Our findings offer insights for potential T cell-focused interventions while underscoring the challenges in eradicating HIV from the CNS, particularly in the context of sub-optimal ART.

R56AI150409 to SSI; RF1AG06001 to JHM),
Foundation for the National Institutes of Health
(HHSN261201500003I to JDL), and Basic
Research Laboratory (75N91019D00024 to JDL).
The funders had no role in study design, data
collection and analysis, decision to publish, or
preparation of the manuscript.

**Competing interests:** The authors have declared
that no competing interests exist.

## Author summary

Antiretroviral therapy (ART) has improved health outcomes of people living with HIV.
However, there are still challenges, especially in the central nervous system (CNS), where
ongoing inflammation can lead to neurological disorders. Our study focused on under-
standing the role of CD4 T cells in the brain during HIV infection and sub-optimal treat-
ment adherence. We used a model with SIV-infected rhesus monkeys to study the AIDS
virus in the brain and surrounding tissues. We discovered that a subset of CD4 T cells,
which are vulnerable to HIV, are present throughout the CNS. During the early stages of
infection, we noticed high levels of the virus in both the brain and nearby tissues. By
examining these CD4 T cells at a single-cell level, we found that they actively respond to
the virus by initiating specific antiviral effector functions to fight it. Overall, our study
helps us understand the role of CD4 T cells within the CNS during both acute and chronic
HIV infection. This knowledge could help us develop new ways to target the virus in the
CNS and devise treatments for complications related to Neuro-HIV.

## Introduction

Improved access to early and sustained antiretroviral therapy (ART) has significantly
enhanced the life expectancy of people living with HIV (PLWH). Studies show that immediate
ART initiation upon diagnosis is the most effective approach to achieving positive long-term
health outcomes [1–4]. However, often, ART is not initiated until well after clinical symptoms
have become evident [5,6]. Delay in treatment initiation leads to systemic viral dissemination
and a decline in CD4 T cell counts, which puts PLWH at a higher risk of developing chronic
inflammatory disorders that can affect multiple organ systems, particularly the brain [7,8].
Indeed, the severity of cognitive impairment at the time of ART initiation is the strongest pre-
dictor of persistent neurocognitive deficits despite long-term ART, underscoring the signifi-
cance of early-stage viral spread and neuroinflammation in the disease process and indicating
that substantial neurological damage is inflicted early following infection [9–12]. As the popu-
lation of PLWH continues to age, gaining a deeper understanding of the immune factors that
drive acute neuroinflammation and contribute to persistent chronic neuroinflammation dur-
ing ART become increasingly vital.

Originally identified as the AIDS dementia complex, HIV-associated neurocognitive disor-
ders (HAND) or Neuro-HIV encompass a range of neurologic complications varying from
mild to severe cognitive and motor impairments [13–15]. While severe forms of Neuro-HIV
are less prevalent today, HIV infection continues to impact the brain, leading to HAND or
HIV-associated brain injury (HABI) [16]. It is hypothesized that early HIV entry into the cen-
tral nervous system (CNS) causes lasting effects on the brain, termed legacy HABI, even with
suppressive ART. Additionally, as the infection progresses, active HABI contributes to a grad-
ual cognitive decline. The role of immune activation in driving neuroinflammation in Neuro-
HIV is crucial, but the specific cellular mediators responsible for both neuroinflammation and
viral persistence in the CNS during chronic infection, as well as immune activation prior to
ART initiation, are not yet fully understood.

The historical identification of multinucleated giant cells in brain parenchyma and enceph-
alitic lesions, along with the association of HIV-1 with brain parenchymal macrophages in
AIDS patients, brought focus to brain-resident myeloid cells as crucial drivers of viral persis-
tence and neuroinflammation [17,18]. Observations of CNS compartmentalized viral variants,

assessed by viral genomic sequencing within the cerebrospinal fluid (CSF) showed ability to mediate infection of cells with lower levels of CD4, and longer half-lives post-acute infection. These data indicated that long-lived cells such as brain-resident macrophages and microglia gradually replace CD4 T cells as the primary source of the virus in the CNS during the chronic phases of infection [19,20]. To reproduce these outcomes, studies in non-human primate models utilized macrophage tropic viral clones that do not depend on high CD4 receptor expression for entry. Additionally, accelerated experimental models intended to increase the frequency, severity, and kinetics of CNS disease in NHP employed CD8 depletion strategies in combination with R5-T cell tropic viruses to induce an immunosuppressive state, leading to the frequent, early development of classic encephalitis lesions typically observed in end-stage Neuro-HIV [21–25]. These accelerated models have contributed to our understanding of how HIV establishes itself in the brain during early infection and quickly multiplies within the CNS during profound immune dysfunction. However, it is essential to note that they do not offer insights into CNS dissemination during most natural transmissions.

To understand the neuroinflammatory triggers leading to neurodegeneration in the modern ART era, it is crucial to leverage our enhanced understanding of CNS immune surveillance to explore viral dissemination in multiple immune-rich niches of the CNS during both acute infection and suppressive ART. To achieve our research objectives, we utilized two macaque cohorts. The first cohort consisted of macaques acutely infected with SIVmac251, and they received no ART. The second cohort was infected with SIVmac251 and assessed up to week 42 post infection. We initiated ART at week 3 and therapy was periodically interrupted to simulate suboptimal adherence, referred to as deferred non-adherent ART regimen. We aimed to characterize the viral-immune interactions involved in establishment and persistence of SIV infection in the CNS. Our findings support three major conclusions. Firstly, we observed that antigen-experienced CD4 T cells present within the brain parenchyma, choroid plexus stroma, dura mater, skull bone marrow, and CSF exhibit a distinctive profile expressing either CCR5 or CCR7. During acute infection, we observed high tissue viral loads within the frontal and temporal lobes, as well as in border tissues, including the lymphoid niche of the skull bone marrow. Associated CD4 T cell depletion within these compartments was suggestive of ongoing CD4 T cell infection in the CNS which was not rescued following sub-optimal ART. Notably, the relative distribution of CCR5 and CCR7 subsets remained stable throughout this process. Secondly, we found that during ART, CSF viral RNA (vRNA) was below the limit of detection (<15 vRNA copies/ml CSF) despite persistent low vRNA levels in the brain, underscoring the intricate nature of viral control in the CNS. Thirdly, we document a previously unknown potential HIV reservoir in the skull bone marrow. Together, our studies provide insights into the interplay between viral dissemination, CNS T lymphocytes, and neuroinflammation, critical to inform the development of targeted approaches to mitigate progressive neurodegeneration in the context of modern ART.

## Results

### Study design

To understand HIV-1 CNS involvement during both the acute and chronic phases of infection, we conducted studies using two macaque cohorts. We employed CCR5-tropic SIVmac251 to replicate acute and chronic HIV-1 effects on the CNS during natural transmission. The first cohort, known as the Acute 251 cohort, comprised four macaques assessed at week 3 pi (SIV, w3). The second cohort, referred to as the Chronic 251 cohort (n = 6), was followed for a duration of up to 42 weeks post-infection (pi). Throughout the study, we collected plasma and CSF samples to capture viral and immune kinetics in both the systemic and the CNS

compartments. Four uninfected age-matched animals served as controls. At necropsy, tissues were collected following trans-cardiac saline perfusion to ensure that the assessment of immune cells in the CNS compartment reflected CNS tissues without contributions from vasculature contents. Detailed information about the animals can be found in **S1 Table**. Gating strategy for identifying T cells in control and SIV CNS tissues is outlined in **S1 and S2 Figs**.

To capture early-stage dynamics in the CNS prior to ART initiation, we utilized the Acute 251 cohort (**Fig 1A**). In this study, we applied both flow cytometry analysis of CNS immune

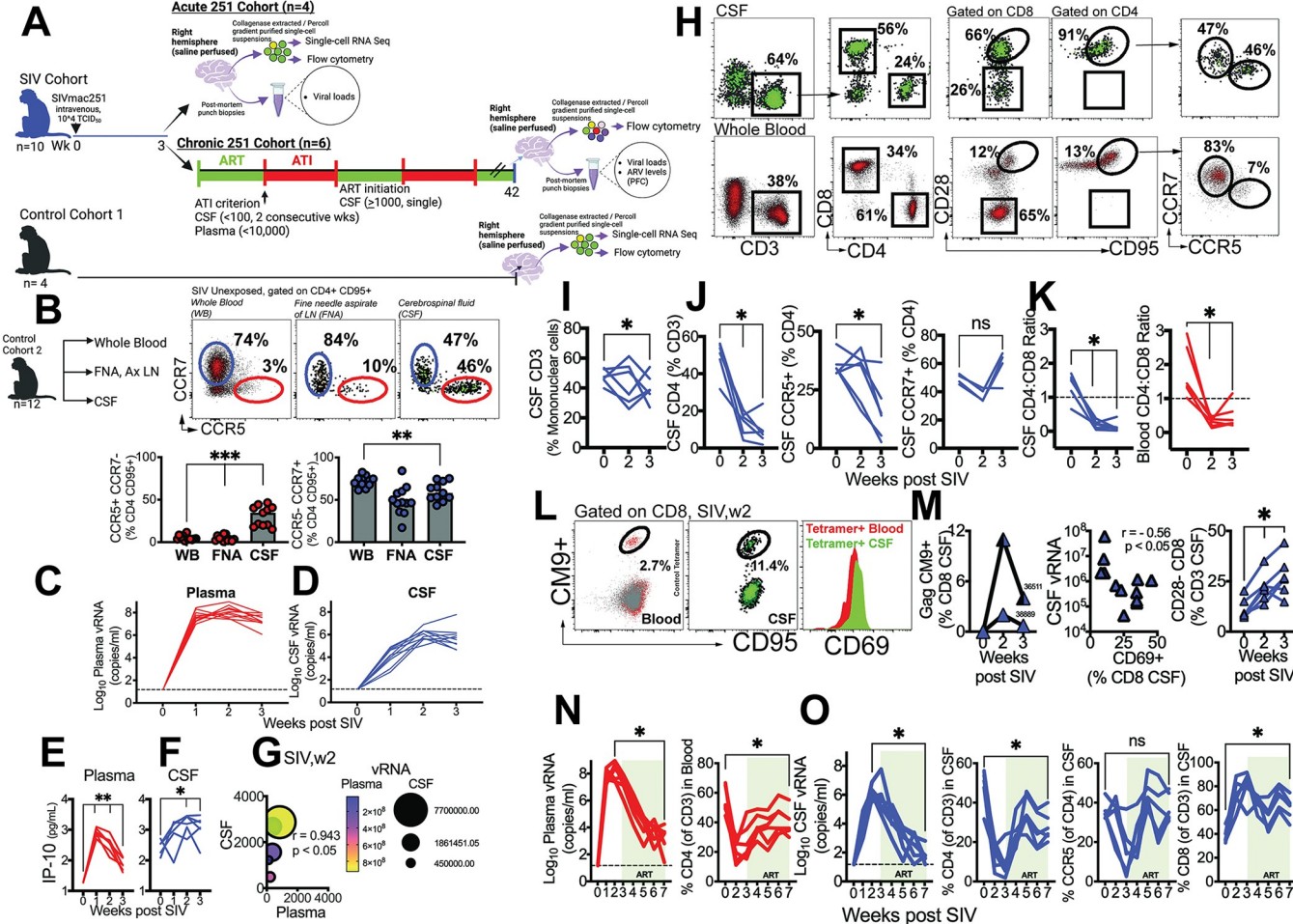

**Fig 1. CNS viral dissemination and neuroinflammation linked to decline in CSF CCR5+ CD4 T cells.** Study design showing (**A**) SIV (Acute 251 (n = 4) and Chronic 251(n = 6) cohorts and Control 1 cohort (n = 4). (**B**) Flow cytometry plots illustrate discrete distribution patterns of CCR5 and CCR7 on CD4+ CD95 + cells in blood, lymph node fine needle aspirate (FNA), and CSF. Bar graphs show frequencies of CD4 subsets across compartments. Animals from Control Cohort 2 assessed (n = 12). Kinetics of viral RNA (copies/mL) measured by RT-qPCR in (**C**) plasma and (**D**) CSF following SIVmac251 infection (n = 10, both acute and chronic 251 cohorts assessed). Horizontal dashed line indicates limit of detection (15 vRNA copies/ml). Concentrations of IP-10 (pg/mL) measured in (**E**) plasma and (**F**) CSF by Legend Plex flow-cytometry based bead assay. (**G**) Bubble plot shows correlation of CSF and plasma IP-10 at week 2 post SIVmac251, Spearman correlation, two-tailed p value shown. Bubble size denotes CSF viral load, and bubble color plasma viral load (n = 6, chronic 251 cohort assessed). (**H**) Flow cytometry plot illustrates CD4+ CD95+ T cells express CCR5 and CCR7 in CSF. (**I**) Kinetics of CSF T cells during acute SIV. (**J**) Kinetics of CD4 T cells in CSF; graphs show % CD4 T cells, %CCR5+ CD4 T cells and % CCR7+ CD4 T cells in CSF. (**K**) CD4:CD8 Ratio in CSF and Blood (n = 6, chronic 251 cohort assessed). (**L**) Flow plot shows Gag CM9+ CD8 T cells in blood and CSF (in A*01 animal 36511). CM9+ CD8 T cells in CSF express CD69. (**M**) Kinetics of CM9+ CD8 T cells in CSF (n = 2 A*01s in chronic cohort). Correlation plot shows CD69+ CD8 T cell frequencies in CSF at weeks 2 and 3 inversely associate with CSF vRNA. Spearman correlation, two-tailed p value shown. Kinetics of CD28- CD8 CSF T cells (% CD3) shown (n = 6, chronic 251 cohort assessed). (**N**) Plasma viral loads during ART (week 3–7), CD4 T cell rebound following viral suppression. (**O**) shows CSF vRNA during ART and CD4 T cell rebound and reconstitution of CCR5+ CD4 T (n = 6, chronic 251 cohort assessed). Significant differences by two tailed Wilcoxon matched-pairs signed rank test, ***, p< 0.01; **, 0< 0.01 in B. Significant differences by one tailed Mann Whitney test, *, p< 0.05 in E and F. Significant differences by one-tailed Wilcoxon matched-pairs signed rank test, *, p< 0.05 in I-O. Schematics were generated using BioRender.

cells and single-cell RNA sequencing on CD45+ cells extracted from the brain. This comprehensive approach allowed us to gain deeper insights into the acute neuroinflammatory programs initiated in CNS CD4 T cells during this critical stage. We collected post-mortem punch biopsies of the brain parenchyma, immediately after the brain was excised, for viral quantification. The remainder of the right hemisphere was collected in media and processed immediately post-harvest to extract single cell suspensions for flow cytometry analysis.

In the Chronic 251 cohort, our objective was to induce cycles of viral suppression and rebound throughout the course of infection, resembling scenarios of intermittent poor adherence to medication or drug resistance, as observed in real-life chronic HIV infections. To achieve this, we employed a 3-drug ART regimen comprising of nucleoside/nucleotide reverse transcriptase inhibitors (NRTIs) Emtricitabine (FTC), tenofovir disoproxil fumarate (TDF) and the integrase inhibitor Dolutegravir (DTG), which was initiated at week 3 post-SIV inoculation, after peak viremia. In clinical studies, FTC and DTG attain therapeutic concentrations within the CNS [26, 27], while TFV's limited CSF diffusion, estimated at only 5%, leads to lower CNS penetration effectiveness scores [28, 29]. As CSF escape is an indicator of resistance, longitudinal assessment of paired plasma and CSF viral loads was performed.

Treatment interruption commenced when CSF viral (v) RNA levels fell below 100 copies/mL at two consecutive time points, with corresponding plasma viral loads below 10,000 copies/mL (ATI: weeks 9–10 post-infection). An exception to this protocol was animal 38359, where ATI was initiated at week 9 when CSF vRNA was 120 copies/mL, a deviation prompted by a decline in plasma viral loads (S2 Table).

Subsequently, ART was re-initiated when CSF vRNA exceeded 1000 copies/mL at a single time point. These repeated cycles of treatment interruption and initiation continued throughout the chronic phase of infection. Two-to-four weeks prior to necropsy, all SIV-infected animals received ART, and levels of antiretroviral drugs in the plasma, CSF, prefrontal cortex (PFC), and colon were quantified.

## CNS viral dissemination and neuroinflammation linked to decline in CSF CCR5+ CD4 T cells

In line with observations in humans [30, 31], we identified that the non-inflamed CSF allows entry of CD4 T cells, typified by an antigen-experienced CD28+ CD95+ phenotype. Notably, we identified the presence of a distinct CCR5+CCR7- subset, comprising approximately 34% (median) of antigen-experienced CD4 T cells in the CSF (range: 15–46%) in contrast to only 4% observed in blood and lymph nodes (Fig 1B). The CCR5+ CCR7- subset of CD8 T cells exhibited higher prevalence in the blood (median 26%). Interestingly, like their CD4 counterparts, these frequencies were even more elevated in the CSF (median 76%, S3A Fig). On the other hand, CCR5- CCR7+CD4 T cells were most prevalent in the blood, constituting 73% (range: 61–82%), whereas they accounted for 58% in the CSF (range: 43–75%).

Based on the presence in the CSF of CCR5+ CCR7- CD4 T cells, consistent with ongoing immune surveillance, we posited that rapid influx of infected CD4 T cells into the CSF following systemic viral replication would lead to acute CNS viral dissemination. Complete blood counts with differential quantitation revealed an expected decrease in CD4 T cell counts in 5/6 animals (4-fold decline at week 4 relative to week 0, p < 0.05) with no significant change in total lymphocyte counts during acute infection (S3B–S3D Fig). During the initial week post-infection, the median levels of CSF vRNA were approximately 15,000 copies. By the second week, CSF vRNA reached a peak of $1.2 \times 10^6$ copies before gradually declining to $4.4 \times 10^5$ copies by the third week. These observed patterns closely paralleled viral kinetics in plasma, albeit at lower levels (Fig 1C–1D).

Measurement of levels of interleukin 8, monocyte chemotactic protein (MCP-1) and interferon protein 10 (IP-10) demonstrated significant induction of MCP-1 and IP-10 in plasma, peaking at week 1 and gradually declining afterward, while remaining significantly higher than baseline levels at week 3 (**Figs 1E and S4**). In contrast to plasma levels, CSF MCP-1 concentrations did not change significantly, and CSF IP-10 levels displayed a distinct pattern, continuously increasing over time (**Fig 1F**). At week 2, CSF IP-10 levels showed a strong correlation with plasma IP-10 levels, indicating that either CSF IP-10 was influenced by systemic induction, or that intrathecal IP-10 production occurred at a similar level as systemic IP-10 induction (**Fig 1G**).

Despite high viral loads, CD28+ CD95+ cells constituted the majority of CD4 T cells in the CSF during acute infection (**Fig 1H**). At week 3, a notable decrease in the proportion of T cells in the CSF was observed (**Fig 1I**), driven by a sharp decline in CD4 T cell frequencies during weeks 2 and 3 (**Fig 1J**). Remarkably, although IP-10, which promotes $T_h1$ cell ingress, was elevated, the proportion of CCR5+ CD4 T cells in the CSF declined. However, the CCR5+ CD8 T cell and CCR7+ CD4 T cell subsets remained relatively stable, suggesting the decline was specific to viral replication in target cells within the CSF (**S5 Fig**). As a result, a significant decrease in CD4:CD8 ratio in both CSF and blood ensued (**Fig 1K**).

Quantifying Gag-specific CD8 T cells using the CM9 tetramer in A*01 animals (n = 2) unveiled the presence of antigen-specific CD8 T cells infiltrating the CSF (**Fig 1L**). These cells prominently displayed CD69 expression, signifying activation. Correspondingly, analysis of CD69+ CD8 T cell frequencies demonstrated inverse association with CSF vRNA levels. Furthermore, an increase in CD28- effector memory CD8 T cells in the CSF was observed, indicative of active CD8 T cell surveillance of the CNS during acute SIV infection (**Fig 1M**).

Viral suppression following ART initiation at week 3 led to significant CD4 T cell rebound both in systemic and CSF compartments (both p < 0.05) with expected reconstitution of CCR5+ CD4 T cells (**Fig 1N–1O**). The decrease in CD4 T cells during viral replication and rebound following viral suppression supports local viral replication as a contributor to CSF viral loads. This interpretation is supported by evidence in macaques and humans of CD4-cell associated spliced vRNA within CSF indicative of active CD4 infection [32]. Since our studies did not assess cell-associated CSF virus, however, we cannot rule out the potential contribution of virions transiting to the CSF from other sites of origin to total CSF vRNA levels.

Assessment of CSF albumin, total protein, and glucose levels indicated that the blood-brain barrier (BBB) maintained its functional integrity despite ongoing viral replication and neuroinflammatory response (**S6 Fig**).

## CCR5+ CD4 $T_h1$ cells populate brain parenchyma

Prior to delving into the effects of infection on brain T cell responses, we elucidated the distribution of CD4 T cells within the non-inflamed CNS of SIV-uninfected macaques. The tissues we analyzed included the brain parenchyma and its border-associated compartments, such as the choroid plexus stroma (ChP). The ChP plays a crucial role as the interface between the circulation and the CSF. Given its immune composition, the ChP was gently extracted from the frontal horn and the body of the lateral ventricle, using clean forceps, with care taken to avoid disruption of surrounding tissue. The extracted ChP was subjected to visual inspection to verify absence of contamination from the surrounding parenchyma and was subsequently pooled together. Additionally, we studied the dura mater (dura), which represents the outermost meningeal layer and supports the meningeal lymphatics. Another important compartment we examined was the skull bone marrow (Sk BM), allowing us to assess the lymphoid niche of the calvaria [33–35]. To complement our investigation, we also included the draining lymph nodes of the brain, the deep cervical lymph nodes (dCLN) and the ileum as a non-CNS tissue.

As demonstrated in **Fig 2A**, flow cytometry analysis of single cell suspensions obtained from CNS tissues of SIV-unexposed control macaques, following saline perfusion, revealed clear identification of distinct CD4 and CD8 T cell subsets. Notably, there was a predominance of CD8 T cells over CD4 T cells in the brain, with a CD8:CD4 ratio of 2.8 (**Fig 2B**). To delineate CD4 T cell helper profiles in the CNS, we examined the brain, Sk BM, and spleen for expression patterns of CCR5 in the context of CCR6, a marker for $T_h17$ cells and CXCR3, the canonical $T_h1$ marker. Our analysis revealed that CD4 T cells in the brain expressed minimal amounts of CCR6 but were predominantly CXCR3+, consistent with CXCR3-mediated ingress of T cells to the CNS. Among CXCR3+ CD4 $T_h1$ cells, the expression of CCR5 was found to be approximately 47% in the brain, 40% in SkBM, and 14% in the spleen. This indicates that a significant proportion of $T_h1$ cells in the brain and SkBM could potentially be susceptible to R5-tropic infection through CCR5-mediated mechanisms (**Figs 2C and S7A–S7B**). The evaluation of integrin heterodimers $\alpha_4\beta_7$ and $\alpha_4\beta_1$ revealed relatively higher $\alpha_4\beta_1$ expression. In a Boolean analysis that considered CCR5 co-expression, the predominant subset consisted of $\alpha_4\beta_1$+ CCR5+ cells, indicating that $T_h1$ cells in the CNS concurrently expressed both $\alpha_4\beta_1$ and CCR5. A comparable expression pattern was also noted in the spleen, as illustrated in **Fig 2D**.

The distinct patterns observed in the CSF for CCR5 and CCR7 were similarly observed within the brain parenchyma and other CNS tissues (**Fig 2E**). However, the frequencies of CCR7+ CD4 T cells were lower in the brain, and on average in the Dura, compared to the CSF (**Fig 2F**). Furthermore, when we examined the polyfunctionality of CD4 T cells, specifically their ability to produce cytokines TNF$\alpha$, IFN$\gamma$, and IL-2 after stimulation with PMA/ionomycin, we observed robust cytokine production in brain CD4 CD95+ T cells, like their counterparts in the spleen (**Fig 2G**). Notably, there were some distinctions; splenic CD4 T cells exhibited a higher propensity to produce IL-2, leading to increased frequencies of IL-2 single-positive and IL-2/TNF$\alpha$ co-producing cells. Conversely, the brain showed significant prevalence of IFN$\gamma$ single-positive cells, aligning with the phenotypic data underscoring functionality of $T_h1$ cells in the brain, akin to their counterparts in lymphoid tissues (**Fig 2H**).

## CD4 T cells depleted in CNS tissues during acute SIV infection

Next, we studied the Acute cohort (**Fig 3A**) and investigated the impact on CD4 T cells in different CNS compartments during the acute phase of infection. Our analysis of all examined CNS tissues revealed a notable decrease in the relative proportion of CD4 T cells, accompanied by a corresponding increase in CD8 T cells compared to uninfected control animals (**Fig 3B**).

During acute SIV infection, the brain and border tissues (dura and Sk BM) displayed distinct phenotypic distributions of CCR5+ and CCR7+ CD4 T cell populations, similar to the distribution seen in the non-inflamed brain during homeostasis. Notably, the CCR5+ CD4 T cell subset exhibited higher expression levels of CD69, PD-1, and CXCR3, indicating an activated and effector-like phenotype, while the CCR7+ CD4 T cells showed a quiescent phenotype, suggestive of a more resting state (**S7C–S7D Fig**). In the CNS tissues, both CCR5 and CCR7 subsets of CD4 T cells were present at varying frequencies. The brain predominantly had CCR5+ CD4 T cells, whereas the spinal cord and Sk BM showed enrichment of CCR7 + CD4 T cells, resembling the distribution observed in bone marrow of long bones and lymphoid tissues (**Fig 3C–3D**).

To investigate the relationship between CCR5 and CCR7 expression during acute SIV infection, we examined the relative co-expression of two markers—PD-1, which indicates TCR stimulation, and CD69, a marker for acute activation and tissue residency. By analyzing t-SNE plots gated on CD4 CD95 T cells expressing a combination of either CCR5, CCR7,

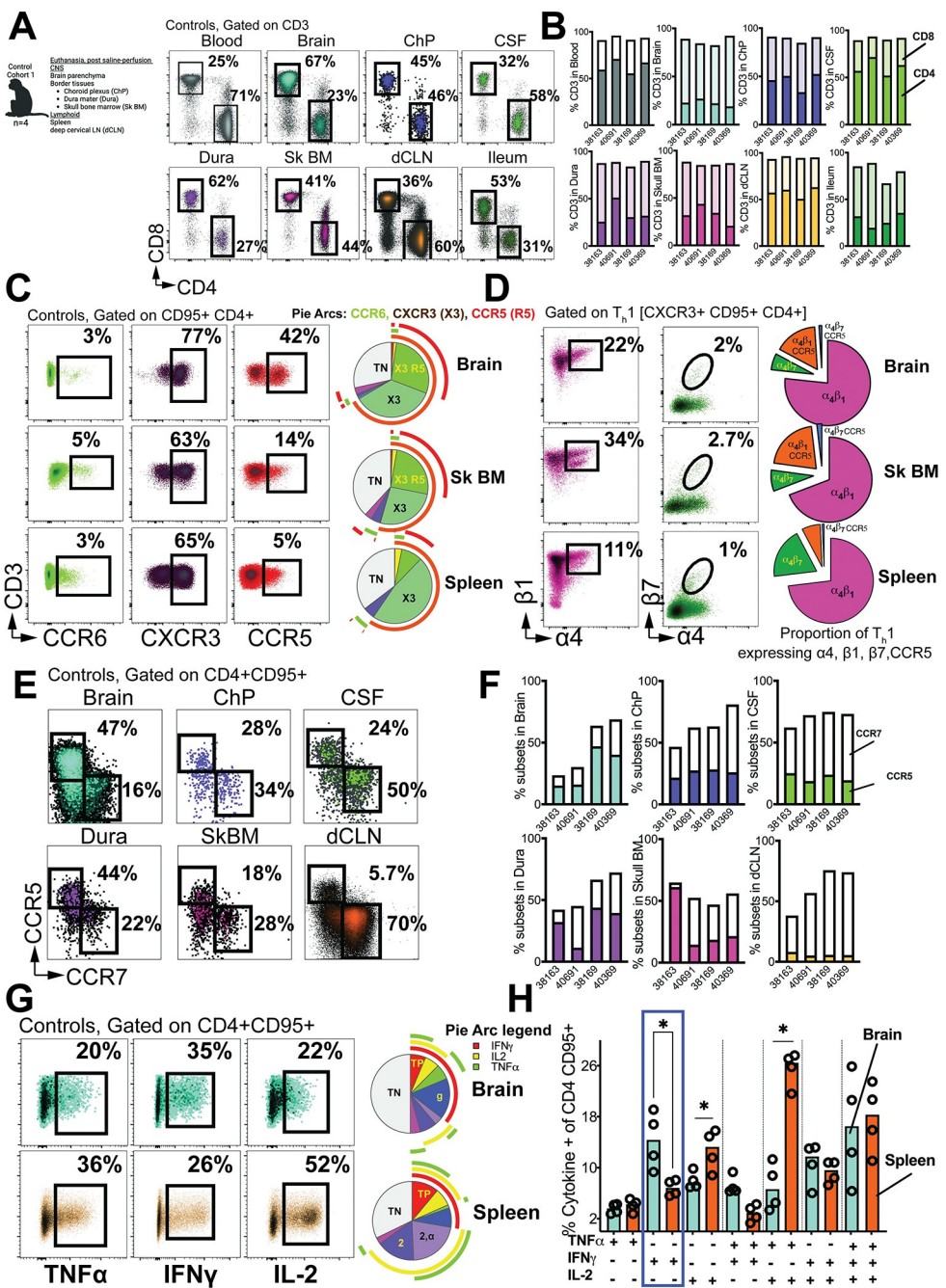

**Fig 2. CCR5+ CD4 T_h1 cells populate brain parenchyma. (A)** Flow cytometry plots illustrate frequencies of CD4 and CD8 T cells in blood, CNS tissues, dCLN, and ileum in controls. **(B)** Contingency plots show % CD4 (solid) and % CD8 (transparent) in each of the control cohort 1 animals (n = 4) assessed. **(C)** Surface expression of CCR6, CXCR3, and CCR5 on CD95+ CD4 T cells. Pie chart shows preponderance of CXCR3+ CCR5+ (X3 R5) subset in brain relative to spleen. **(D)** Flow plots and pie chart illustrate expression of $\alpha_4\beta_1$, $\alpha_4\beta_7$, and CCR5 on CD4 T_h1 cells. **(E)** CCR5 / CCR7 distinction in CNS. **(F)** Contingency plots show % CCR5 (solid) and % CCR7 frequencies (transparent) in each of the control cohort 1 animals (n = 4) assessed. **(G)** Flow plots show cytokine production following 3-hour stimulation with PMA/Ionomycin in controls (n = 4). **(H)** Pie Chart and bar graph show proportion of cytokine producing cells across brain (n = 4) and spleen. (n = 4) Significant differences by one tailed Mann Whitney test, *, p< 0.05 in H. Schematics were generated using BioRender.

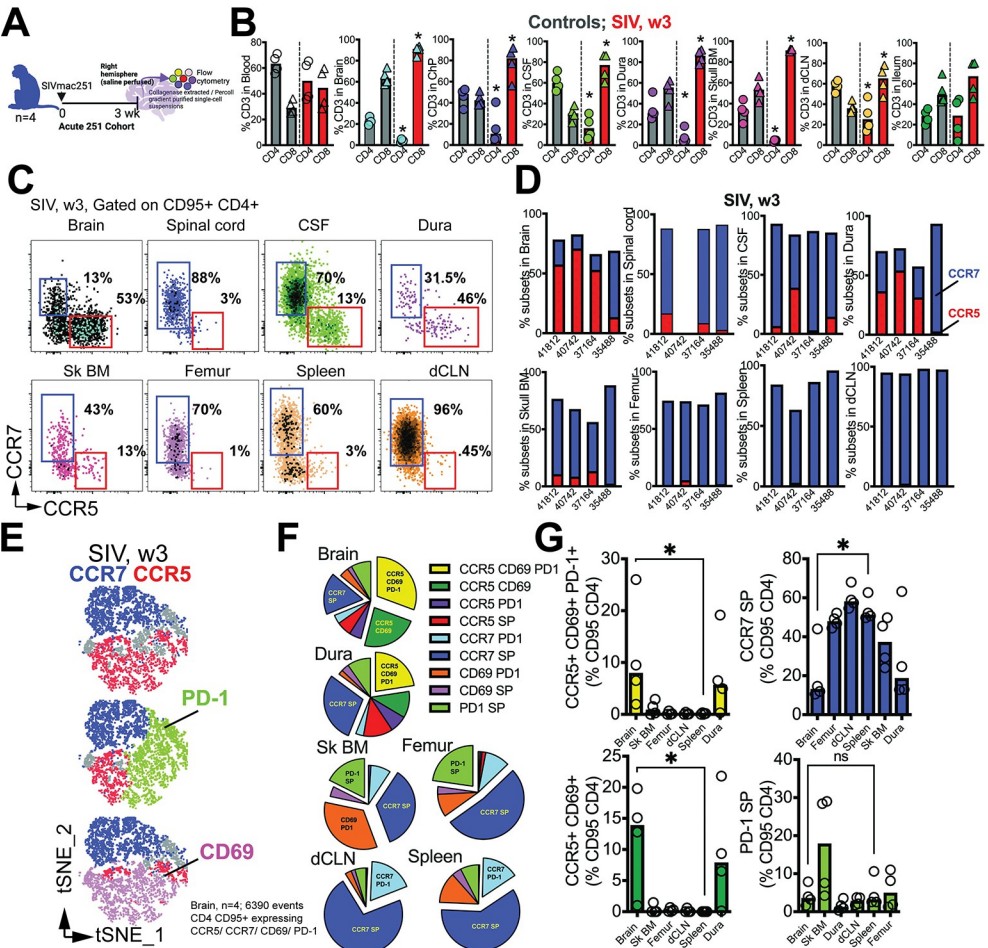

**Fig 3. CD4 T cells depleted in CNS tissues during acute SIV infection.** (A) Acute 251 cohort (n = 4) assessed. (B) Bar graphs show T cell frequencies in blood, CNS tissues, dCLN, and ileum in control and SIV infected (week 3 pi) macaques. (C) Flow plots and (D) contingency plots show CCR5+ / CCR7+ CD4 T cell subset frequencies at 3 weeks post SIV (n = 4). (E) t-SNE plots gated on CD4+CD95+ cells in Brain expressing CCR7/CCR5/PD-1/CD69 (n = 4 samples, 6390 events). (F) Pie chart demonstrating proportion of CD4 T cells expressing combination of markers (CCR7/CCR5/PD-1/CD69). (G) shows distribution of specific subsets across tissues. Significant differences by one tailed Mann Whitney test, *, p< 0.05 in A and F. Schematics were generated using BioRender.

CD69, PD-1, we found that the majority of PD-1 and CD69 expression occurred within the CCR5-expressing CD4 T cells (Fig 3E). Relative distribution of these subsets demonstrated that cells co-expressing CCR5 CD69 and PD-1, as well as CCR5 and CD69, were most abundant in the brain and comparable to frequencies in dura (Fig 3F–3G), relative to other compartments. Conversely, cells expressing CCR7 but neither of the other three markers (CCR7 single positive, SP) were significantly higher in the lymphoid tissues, as expected, relative to the brain. Altogether, the data show distinct compartmentalization of CCR5+ and CCR7 + CD4 T cells and reveal that despite CD4 T cell depletion within the brain parenchyma, both CCR5 and CCR7 subsets co-exist during acute infection.

## T cell clusters within SIV-Infected brain

To gain deeper insights into the inflammatory programs induced in T cells within the brain following SIV infection, we conducted single-cell transcriptomic profiling of CD45+ cells

extracted from the brain of the Acute cohort. This analysis included 4 samples from acute SIV-infected macaques and 2 uninfected controls (**Fig 4A**). For the uninfected control samples, we observed a median of 115,884 mean reads per cell, with a median of 29,278 total genes detected. The SIV-infected brain samples exhibited 107,659 reads per cell and 29,541 identified genes. Following cluster analysis of the SIV-infected brain samples, we identified six distinct immune clusters. Interestingly, all clusters except for the dendritic cell (DC) cluster were shared with control samples reflecting differential DC dynamics induced by SIV infection (**Fig 4B**).

To characterize these cell clusters further, we analyzed highly expressed genes within clusters across all animals. In the CD8 $T_{CM}$ cluster, we observed significantly higher expression of key transcription factors, such as ID2 (1.5-fold relative to $T_{EM}$ in both control and SIV) and JUNB (2.2-fold relative to $T_{EM}$ in control, 1.9-fold relative to $T_{EM}$ in SIV). Additionally, we observed higher expression of essential regulators of T cell signaling, like THEMIS (2.5-fold relative to $T_{EM}$ in control, 1.7-fold relative to $T_{EM}$ in SIV) and BTG1, a regulator of quiescence (1.9-fold relative to $T_{EM}$ in control, 1.8-fold relative to $T_{EM}$ in SIV), among others (**Figs 4C and S8A**). Conversely, the CD8 $T_{EM}$ cluster exhibited the expression of transcription regulators linked to effector differentiation, specifically IKZF2 and ZEB2, both exhibiting an approximate two-fold increase in control samples. Intriguingly, in both control and SIV conditions, over 50% of cells in both CD8 $T_{EM}$ and $T_{CM}$ clusters expressed CD69, highlighting the presence of a diverse spectrum of differentiation states within these clusters. Furthermore, in control samples, we observed an enrichment of IL7R in the CD8 $T_{CM}$ cluster, while this enrichment was absent in the SIV samples, signifying transcriptional changes in response to infection.

The CD4 $T_{CM}$ cluster displayed expression of canonical $T_{CM}$ genes CD28 and IL7R, while the monocyte/macrophage cluster showed FC receptor and Class II gene expression. The NK cell cluster was marked by expression of killer cell lectin-like receptor (KLR) genes [36]. Notably, the DC cluster was unique to the SIV condition and showed distinct expression of the canonical myeloid transcription factor IRF8, as well as genes regulating antigen presentation, including CD74 [37].

Next, we aimed to identify viral transcripts within individual cells and assess their correlation with CD4 T cell immune clusters. To quantify viral transcripts, we developed a custom reference by integrating the SIV isolate SIVmac251.RD.tf5 (SIVmac251) data with the *M. mulatta* (Mmul_10) genome files. Once our tailored reference was successfully established, it served as the basis for the count matrix, which included viral transcripts, facilitating subsequent analyses. Within SIV-infected brain and spleen samples, we observed a spectrum of SIV RNA transcript expression levels, whereas in the control samples, no expression was detected, confirming the validity of our approach (**S8B**). Following the application of a threshold for expression above 1, we observed majority of vRNA-positive cells were concentrated within the CD4 and monocyte clusters in both the brain and spleen with some evidence for background signal in myeloid and CD8 T cell clusters (**S8B–S8C**). Therefore, we subsequently merged CD4 and monocyte clusters. The combined UMAP, with overlayed SIVmac251 reference sequence, highlighted mapped reads associated with CD4 T cell clusters (total number of vRNA+ cells >1 = 11) and, to a certain extent, monocyte/macrophage clusters (total number of vRNA+ cells = 1) in the brain (**Fig 4D–4E**). Examination of splenic CD45+ cells exhibited SIV RNA co-localization with CD4 T cells (total number of vRNA+ cells = 4). The inset displays the percentage of SIV transcripts expressed in vRNA+ CD4 T cells in both the brain and spleen. Microscopic analysis of the prefrontal cortex (PFC) using RNAscope confirmed our PCR data, revealing the co-localization of viral RNA (vRNA) with CD3+ cells in the CNS parenchyma (**Fig 4F**).

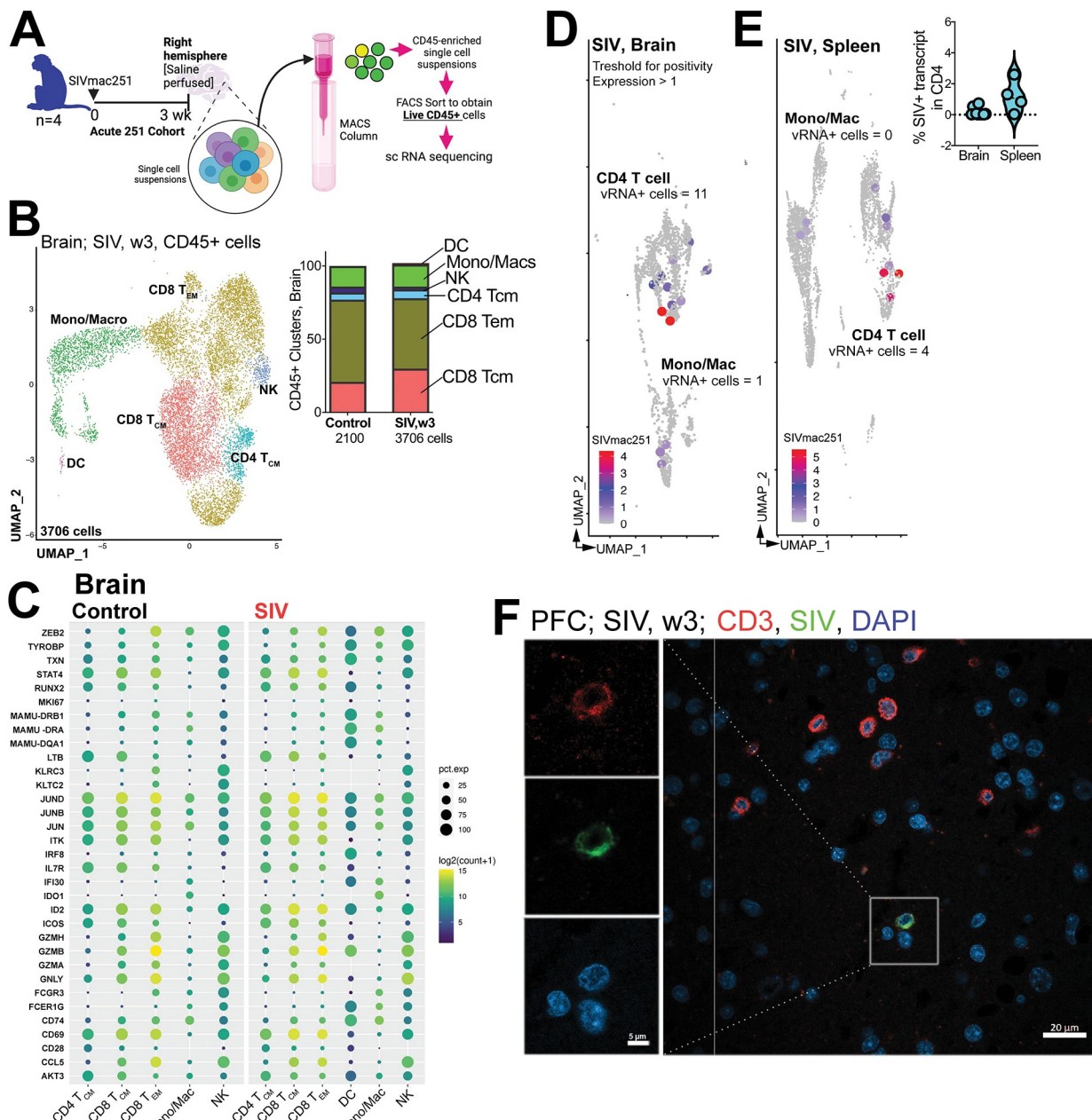

**Fig 4. T cell clusters within SIV-Infected brain. (A)** Schematic of single cell profiling of CD45+ cells in brain (n = 4 Acute 251 cohort, n = 2 Control cohort 1). Sequence alignment to *M.mulatta* (Mmul_10) reference using 10X Genomics protocol (CellRanger V.6.0) was performed. The generated cell-by-gene count matrix was used for downstream analysis using the Seurat based integrative analysis workflow. The filtered count matrix (percentage of mitochondrial reads <10, and gene expressed in more than 10 cells) was log-normalized, with top variable genes used for graph-based cell clustering with a resolution of 0.5 and visualized using Uniform Manifold Approximation and Projection (UMAP). **(B)** UMAP of scRNA-seq transcriptional profiles from brain shows 6 clusters. Cell clusters are color-coded based on cell types. Cluster identity was assigned by a combination of approaches—cluster-specific differentially expressed genes, expert knowledge, canonical list of marker genes, and automated annotations using immune reference atlas through SingleR. Inset shows cell proportions in each cluster by experimental group. **(C)** Dot plot of select marker gene expression. Dot size represents proportion of cells expressing gene and color designates expression level. To quantify viral transcripts, we designed a custom reference using CellRanger mkref pipeline. We integrated FASTA and GTF files of SIVmac251 into *M. mulatta* (Mmul_10) genome references. This tailored reference facilitated downstream analysis by including viral transcripts in the count matrix. UMAP of SIV RNA expression in cell clusters (SIV RNA+ cell size increased for clarity) in **(D)** brain and **(E)** spleen. Number of cells from each cluster positive for vRNA provided. After filtering cells expressing SIV transcript above a threshold 1, %SIV+ for CD4 T cells was determined by dividing the count of the SIV transcript by the total gene count (inset). **(F)** shows SIV RNA in parenchyma and perivascular regions of the brain using ISH with probe against SIV RNA. SIV RNA+ (green) CD3+ T cells (red) with nucleus (DAPI, blue) in PFC; box (CD3+ SIV+ cell). Schematics were generated using BioRender.

## T cell effector molecular programs induced within the SIV-Infected brain

Upon comparing each immune cluster between SIV-infected brain samples and controls (**S9 Fig**) and focusing on genes differentially up or downregulated with adjusted p values of < 0.05, we noted the induction of a transcriptional program regulated by interferons, viral infection, and pattern recognition receptors across all clusters (**Fig 5A**). The observed transcriptional program encompassed several key components, such as interferon alpha inducible protein 27 (IFI27, up 8.8-fold, CD4 $T_{CM}$; 6.4-fold, CD8 $T_{CM}$; 7.8-fold, CD8 $T_{EM}$; 13.7-fold, Mono/Mac; 21-fold, NK), IFI16 (5.1-fold, CD4 $T_{CM}$; 3.4-fold CD8 $T_{CM}$; 4.4-fold, CD8 $T_{EM}$; 8.5-fold, Mono/Mac; 9.7-fold, NK); interferon stimulated genes which induce transcription of antiviral factors—ISG15 (2.8-fold CD4 $T_{CM}$; 2.1-fold CD8 $T_{CM}$; 2.6-fold, CD8 $T_{EM}$; 3.6-fold, Mono/Mac; 4.2-fold, NK) and ISG20 (1.6-fold, Mono/Mac [38], and interferon induced protein with tetratricopeptide repeats which mediate molecular signaling by forming complexes with cellular and viral proteins IFIT2 (1.6-fold, Mono/Mac), IFIT3 (1.6-fold, Mono/Mac [39]. Also induced were the MX Dynamin like GTPases 1 and 2 (MX1 (1.8-fold, CD4 $T_{CM}$; 1.4-fold, CD8 $T_{CM}$; 3.3-fold, Mono/Mac; 2.3-fold, NK), MX2 (2.4-fold, Mono/Mac; 1.5-fold, NK). Notably, within the monocyte/macrophage cluster the cytidine deaminase targeting primate lentiviruses, apolipoprotein B mRNA editing enzyme catalytic subunit 3A (APOBEC3A), was induced 3-fold [40].

The pattern of antiviral gene expression in brain parenchymal CD4 T cells closely resembled that observed for CD4 and monocyte/macrophage clusters in the spleen, suggesting that CD4 T cells in both compartments exhibited similar biological responses to viral infection (**Fig 5B**). Within CD4 $T_{CM}$ cluster, genes representing pathways regulating antiviral response, cytolytic function, defense response, metabolism were highly induced (**Fig 5C**). Among genes significantly downregulated in the CD4 T cell clusters within the SIV-infected brain and spleen, we observed the following changes: the S-adenosylmethionine sensor, BMT2, exhibited substantial reductions, with a remarkable 22-fold decrease in the brain and an even more pronounced 25-fold decrease in the spleen. The aryl hydrocarbon receptor (AHR) was also downregulated (1.6-fold in the brain and 2.9-fold in the spleen), along with the RNA binding protein RBPMS (3.2-fold in the brain and 1.8-fold in the spleen). Moreover, the anti-inflammatory adenosine receptor ADORA2b (12-fold in the brain and 13-fold in the spleen) and the glucocorticoid receptor NR3C1 (1.7-fold in the brain and 1.6-fold in the spleen) were decreased. Notably, the downregulation of IL7R (1.6-fold) and CD4 (1.4-fold) in brain CD4 T cells indicated T cell activation (**Fig 5D**).

Based on induction of genes related to T cell activation and differentiation (CCL5, LAG3, ZEB2) and cytolytic function (GZMM, NKG7), and downregulation of IL7R in the $T_{CM}$ cluster during SIV, we hypothesized that the observed changes in gene expression patterns reflected a spectrum of T cell differentiation states. To test this hypothesis, we utilized an unbiased pseudotime approach where we identified four distinct lineages (**S10 Fig**), each characterized by unique gene expression profiles. Lineage 4 predominantly consisted of CD8 $T_{EM}$, while Lineages 1 and 3 distinguished themselves through the expression of genes indicative of the transition from $T_{CM}$ to $T_{EM}$ states (**Figs 5E and S10B–S10C**). Along this trajectory was induction of genes associated with cell cycle progression (TK1, MKI67, EIF1, S100A10, S100A4), immune cell activation and differentiation (ZEB2, KLF2, CD52) [41], cytotoxic function (PFN1, GZMB, GZMH, NKG7, and CST7) [42]. Modest upregulation of genes from the ribosomal family, involved in regulating translation was also observed. In contrast, canonical $T_{CM}$ genes, such as IL7R and LTB, were downregulated in this lineage, suggesting a distinct pattern of gene expression associated with the differentiation process induced by SIV infection. **Fig 5F** illustrates gene expression changes across pseudotime in T cells from control and SIV brain.

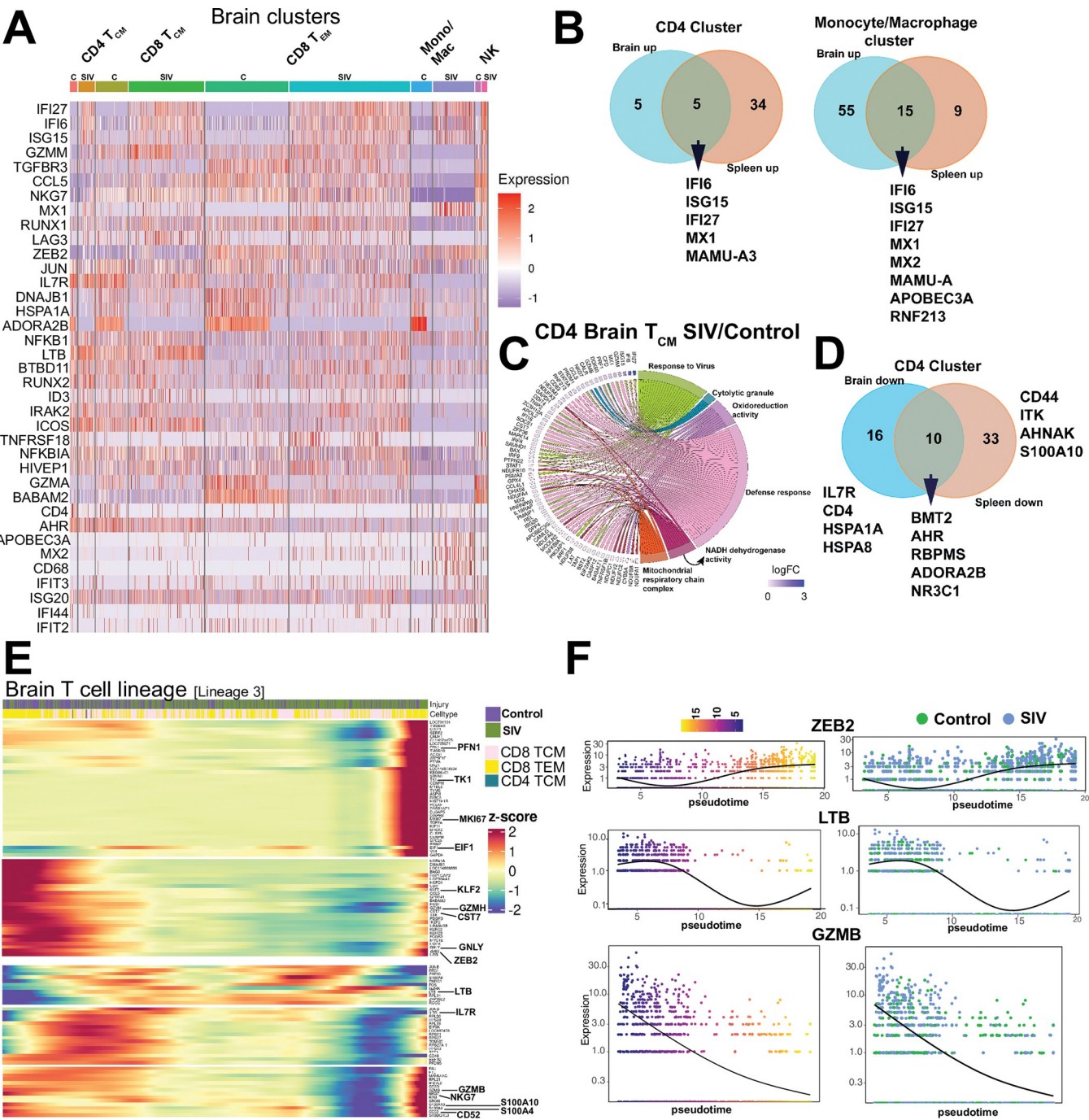

**Fig 5. T cell effector molecular programs induced within the SIV-Infected brain.** Differential gene expression (DGE) analysis of the immune clusters across conditions was performed using functions from Seurat; selection threshold of (adjusted p-value < 0.05, |log2 FC| > 0.25) based on Benjamini-Hochberg correction. **(A)** Heat map of DGE genes in controls (C) versus SIV for each immune cluster. **(B)** Venn diagram shows shared interferon stimulated genes upregulated post SIV across brain and spleen CD4 T cell and monocyte/macrophage immune clusters. **(C)** Chord plot show pathways and corresponding genes enriched in SIV versus control CD4 T$_{CM}$ cell cluster in brain. **(D)** Venn diagram shows shared genes downregulated post SIV in brain and spleen CD4 T cell clusters. We used the monocle3 based workflow to estimate lineage differentiation between the cell populations based on the experimental conditions. We extracted the subsets of identified cell types from our integrated Seurat object and further inferred the trajectory graphs. Using the defined root node (T$_{CM}$), we chose lineages based on the shortest path connecting the root node and the leaf node. After establishing different lineages, we implemented a differential gene test to find genes that changed as a function of pseudotime based on a combination of Moran's statistic and q-value and visualized using heatmaps and individual gene trajectory plots. **(E)** Heatmap (Lineage 3) shows changes in gene expression in lineage comprising of T cells. Along this trajectory was induction of genes associated with cell cycle progression (TK1, MKI67, EIF1, S100A10, S100A4), immune cell activation and differentiation (ZEB2, KLF2, CD52) [41], cytotoxic function (PFN1, GZMB, GZMH, NKG7, and CST7). Canonical T$_{CM}$ genes, such as IL7R and LTB, were downregulated in this lineage.

**(F)** shows expression levels genes of select genes from heat map (ZEB2, LTB, GZMB) along pseudo-time as a function of infection. Schematics were generated using BioRender.

Notably, increasing expression of ZEB2 aligns with infection induced effector T cell differentiation Conversely, the decline in LTB expression is linked to SIV indicative of T cell activation, while lower GzmB expression in control T cells aligns with induction of cytolytic programs following SIV infection. Collectively, the data revealed induction of an antiviral transcriptional program across all immune clusters, underscoring the robustness of the immune response; each cluster exhibited unique inflammatory pathways tailored to complement the specialized antiviral functions of individual immune subsets.

## vRNA in brain regions controlling cognitive function and within CNS border tissues

The sc RNA seq and flow cytometry data demonstrating co-existence of activated CCR5 + CCR7- CD4 T cells alongside quiescent CCR7+ CCR5- CD4 T cells in the Acute cohort raised possibilities regarding their potential roles in viral replication. To delve into this hypothesis, we quantified cell-associated vRNA and vDNA levels within specific CNS tissues (**Fig 6A**). We collected post-mortem punch biopsies (~30 mg) from specific regions of interest, including both white (w) and gray (g) matter regions of the PFC and temporal lobe, such as the superior temporal sulcus (STS). Additionally, we assessed other CNS regions, including the hippocampus (Hp), pituitary (Pit), a circumventricular organ, as well as the border tissues (ChP, dura, Sk BM). The data revealed high levels of vRNA within the brain parenchyma, with a median of 0.16x10^5 vRNA copies per 10^6 cells (**Fig 6B**). The viral loads in the border tissues were also high, with the ChP exhibiting a median of 5x10^5 vRNA copies, the Dura

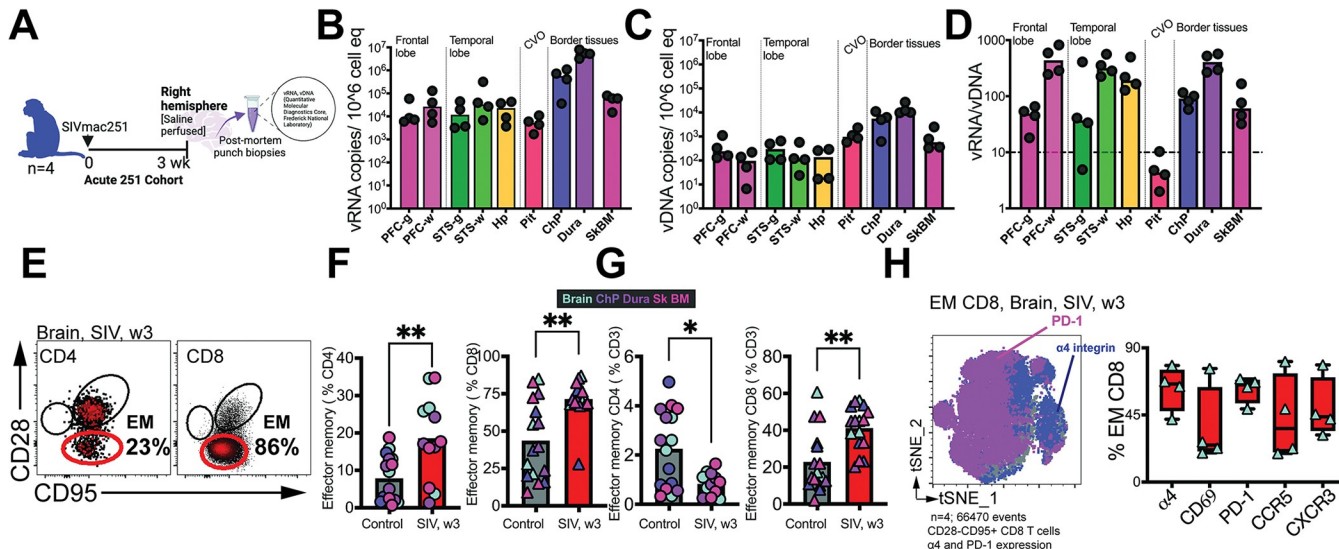

**Fig 6. vRNA in brain regions controlling cognitive function and within CNS border tissues. (A)** Acute 251 cohort (n = 4) assessed**. (B)** SIV vRNA **(C)** SIV vDNA (copies/10^6 cells) in brain regions specified (RT-qPCR on post-mortem punch biopsies from specified regions. **(D)** shows vRNA/vDNA ratio. **(E)** Flow plot of CD4 and CD8 T cells in brain parenchyma during SIV shows identification CD28- CD95+ effector memory (EM) cells **(F)** increase in % EM CD4 and CD8 T cells during acute SIV in CNS tissues specified. **(G)** shows decrease in EM CD4 (% CD3) and increase in EM CD8 (% CD3) T cells during acute SIV in CNS tissues specified. **(H)** t-SNE plots gated on EM CD8 T cells in Brain (n = 4 samples, 66470 events) overlayed with PD1 and α4 integrin expression. Box and whisker plots show expression of specific markers in EM CD8 T cells in brain. Significant differences by two-tailed Mann Whitney test, **p< 0.01, * p< 0.05 in F-G. Schematics were generated using BioRender.

5x10^6 vRNA copies, and the SkBM with 0.6x10^5 vRNA copies, indicating widespread viral dissemination throughout the CNS.

Assessment of vDNA across these regions showed a median of 135 vDNA copies/$10^6$ total cell equivalents observed in the frontal and temporal lobes (**Fig 6C**). Viral DNA was prominently observed in border tissues, with the Dura showing the highest levels at 11,000 copies/$10^6$ total cell equivalents. The elevated viral burden in the dura aligns with its role in draining antigens from the CNS. The computation of the vRNA/vDNA ratio indicated active viral expression across all regions within CNS. Particularly noteworthy was the observation of higher ratios in white matter regions, known to harbor T cells [43] (**Fig 6D**). This observation aligns with imaging studies in PLWH, which have demonstrated widespread and rapid loss in white matter volume during the early stages of infection, while loss in gray matter is more defined and localized to specific regions of the parenchyma, such as the caudate nucleus [44, 45]. Collectively, the combination of phenotypic data, sc analysis, RNAscope, and viral load analysis indicates that the CNS is permissive to R5-T cell tropic viruses.

Assessment of T cell phenotype furthermore revealed a significant increase in the relative proportion of the CD28- effector memory subset within both CD4 and CD8 T cells in the brain parenchyma during acute SIV (**Fig 6E–6F**). However, when expressed as a percentage of CD3, a contrasting trend emerged: a decrease in CD28- CD95+ CD4 T cells alongside an increase in CD28- CD95+ CD8 T cells was evident, implying potential CD4 depletion coupled with CD8 T cell infiltration (**Fig 6G**). This observation aligns with trend for inverse association between CCR5+ CD4 T cell frequencies in CNS and concurrent plasma viral load and substantial expression of α4 integrin and PD-1 on CD28- CD95+ CD8 T cells, indicating influx of antigen-stimulated/specific CD8 T cell to the brain parenchyma (**Fig 6H**).

## Decrease in vRNA in brain during antiretroviral therapy

To complement our data characterizing the establishment of CNS infection over the first 3 weeks of infection in the Acute cohort, we turned to the Chronic cohort, in which we initiated ART at week 3 pi (**Fig 7A**). The time taken to achieve viral suppression in the CNS (defined as CSF vRNA copies at or below 15) varied between 3 to 7 weeks (**Fig 7B**). During this period, transient on-ART increases in the CSF vRNA level occurred between 5 and 7 weeks in some animals, except for one animal, 38889 (Mamu A*01). Remarkably, 38889 demonstrated notable CSF viral suppression even without ART suggestive of SIV-specific CD8 T cell mediated viral control in the CSF in this animal, an observation supported by CSF influx of Gag-CM9 CD8 T cells as shown in Fig 1L.

In all animals, there was an initial rapid decay of CSF viremia up to week 6, which closely mirrored the decay of plasma viral loads during ART. Subsequent suppressive periods showed shorter intervals, consistent with lower initial CSF viral loads (median 5,150 vRNA copies/mL) and rapid decay kinetics. Once cycles of ART withdrawal were initiated, plasma and CSF vRNA rebound was observed for all animals, except for 38889, which achieved suppression of CSF vRNA (but not plasma vRNA) while off ART.

To assess evidence of viral expression in the setting of short-term ART suppression of viral replication, we implemented a strategy where all animals were on ART for a period of 2–4 weeks before necropsy. This intervention led to significant reduction of CSF vRNA levels to < 15 copies/mL. We also evaluated the penetration of antiretroviral drugs (ARVs) into CSF. For this purpose, we quantified ARV levels in both plasma and CSF samples obtained at the time of necropsy. To gain a deeper understanding of ARV metabolism and intracellular levels of active metabolites, we investigated the presence of di and tri-phosphorylated forms of TFV/ FTC within tissues. The timing of sample collection was carefully coordinated to measure

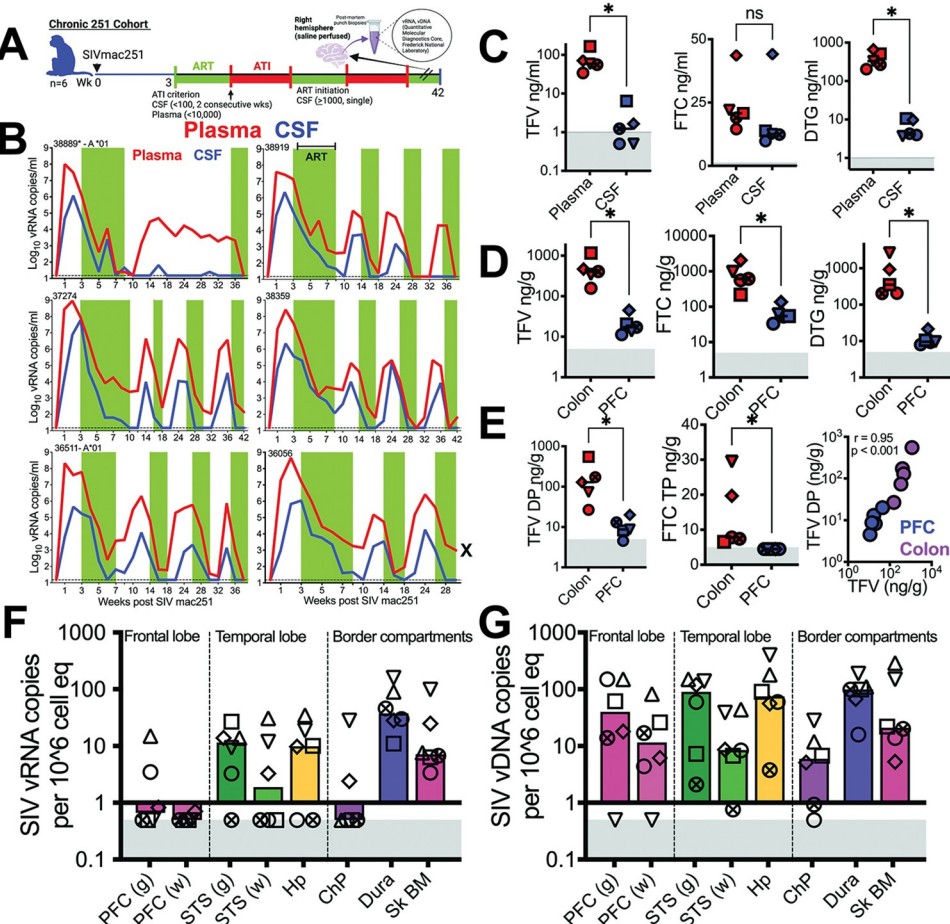

**Fig 7. Decrease in vRNA in brain during antiretroviral therapy. (A)** Chronic 251 cohort (n = 6) assessed. **(B)** Kinetics of plasma (red lines) and CSF (blue lines) viral suppression and rebound (vRNA copies/mL fluid, measured by RT-qPCR) over the course of ART initiation and interruption. Green bars indicate periods of ART with FTC, TDF, and DTG. Horizontal dashed line indicates limit of detection (15 vRNA copies/ml). **(C)** Concentration of ARVs (ng/mL) in plasma and CSF quantified by LC-MS. **(D)** Concentration of ARVs (ng/mg) in PFC and colonic tissue. **(E)** shows active phosphorylated forms of TFV and FTC. Spearman correlation, two-tailed p value shown. Sampling was performed 2–4 weeks post ART initiation with last ARV dose administered 9–12 hours prior to necropsy, FTC = emtricitabine, TDF = tenofovir disoproxil fumarate, DTG = dolutegravir, Gray shaded area represents lower limit of quantification of assay. **(F)** SIV vRNA **(G)** SIV vDNA (copies/10^6 cells) in brain region (RT-qPCR on post-mortem punch biopsies from specified regions. Gray shaded area represents viral loads below threshold of detection. Significant differences by two tailed Wilcoxon matched-pairs signed rank test, * p< 0.05 in C-E. Schematics were generated using BioRender.

trough levels in plasma and CSF. Specifically, samples were obtained 9–12 hours after the last ART dosage and were immediately processed to preserve sample integrity. To examine tissue-specific effects, post-mortem punch biopsies were collected from the PFC and colon, weighed, and flash-frozen for subsequent analysis.

All three ARVs were measured, and the active metabolites of the NRTIs were quantified in colonic (a site of peripheral viral replication) and PFC (a cognitive area of the brain affected in chronic HIV) tissue. Consistent with previous studies [46], FTC exhibited the highest penetration, with CSF levels similar to those observed in plasma (median ng/mL; plasma: 20.8; CSF: 12.5) (**Fig 7C**). DTG was also found above the lower limit of quantification (LLOQ) in CSF for all animals (median ng/mL; plasma: 329; CSF: 4.2, p<0.05). On the other hand, TFV showed

detectable levels in the CSF of only 2 out of 5 animals (median ng/mL; plasma: 58.8; CSF: 1.2, p<0.05). We investigated the penetration of ARVs into brain tissue and observed that all ARVs were able to penetrate the brain, with levels of TFV, FTC, and DTG above the LLOQ (**Fig 7D**). However, it's important to note that these levels were at least 10-fold lower compared to those observed in colonic tissue. Importantly, we observed the accumulation of TFV di-phosphate (DP) in the PFC, indicating the penetration of ARVs into the CNS (**Fig 7E**).

Measurement of CSF viral loads demonstrated that vRNA levels were low to below thresh-old of detection. In the gray matter of the PFC, measurable vRNA levels were observed in only 2 out of 6 animals, while all animals tested negative in the white matter of the PFC (**Fig 7F**). Interestingly, in the temporal lobe, there was a higher likelihood of focal viral expression, with most animals showing vRNA in the STS and Hp. Within the border tissues, detectable vRNA in the dura and Sk BM was found in all animals; however, vRNA in ChP in most animals was not measurable, in line with concurrent absence of vRNA in CSF. Evidence of viral infection was observed with vDNA demonstrable across the brain parenchyma in all animals (**Fig 7G**). These collective findings strongly support the conclusion that lymphotropic viruses establish viral reservoirs in the CNS. Having observed the active induction of IP-10 during acute viral infection within the CNS, we further explored its potential as a biomarker for ongoing viral replication during the chronic phase. Despite viral suppression, we observed a trend for higher IP-10 levels in CSF relative to baseline and concurrent plasma levels at week 7. The lack of a consistent pattern in CSF IP10 during ART and interruption periods diminishes its reliability as a potential CSF biomarker for viral control and rebound during chronic infection (**S11 Fig**).

## Persistent CD4 depletion in CNS during chronic infection

The presence of vRNA in the brain parenchyma and associated border tissues during subopti-mal ART led us to hypothesize that CD4 T cells would remain depleted in the CNS during chronic infection. To explore this, we assessed CD4 T cell frequencies in various CNS com-partments, including blood, brain, choroid plexus, CSF, dura, Sk BM, and dCLN (**Fig 8A–8B**). Our analysis confirmed our hypothesis, revealing a significant reduction in CD4 T cell fre-quencies compared to uninfected control animals (**Fig 8C**).

In humanized mice, HIV infection leads decrease in CD4 T cell counts and an increase in CD8 T cell counts in the brain [47]. However, after the initiation of ART treatment, CD4 T cell levels in the brain return to normal, a contrast to current observations in macaques. This dis-parity may be attributed to several factors, such as the multiple rounds of ATI and the rela-tively brief duration of ART in macaques. Furthermore, CD4 T cell levels are reported as percentages herein, a metric influenced by both CD4+ and CD8+ T cell counts.

To further investigate the changes within CD4 T cell subsets during acute and chronic infection in the brain parenchyma, we focused on $T_h1$ cells expressing $\alpha_4\beta_1$ and CCR5, and compared them to distribution observed in the spleen. The proportion of CXCR3+CCR5 + CD4 T cells remained constant with infection, while there was a relative decline in these cells in the spleen during the acute phase. Meanwhile, both $\alpha_4\beta_1$+ CD4 $T_h1$ and $\alpha_4\beta_1$+ CCR5+ $T_h1$ cells were significantly reduced during acute infection in both the brain and spleen (**Fig 8D**).

During the chronic phase, subsets expressing either CCR5 or CCR7 were still observed (**Fig 8E**). t-SNE plots of CD4 CD95 T cells expressing CCR5/CCR7/CD69/PD-1 demonstrated a distribution pattern similar to that observed during acute infection (**Fig 8F–8G**). Notably, cells expressing PD-1 but not CCR7, CD69, or CCR5 (PD-1 single positive (SP) were present at higher frequencies during the chronic phase (**Fig 8H**). This increase in the relative frequencies of the PD-1 SP subset was also observed in the dCLN. Furthermore, while PD-1+ CD69+ CD4 T cells were not significantly higher in the brain during the chronic phase, their frequencies

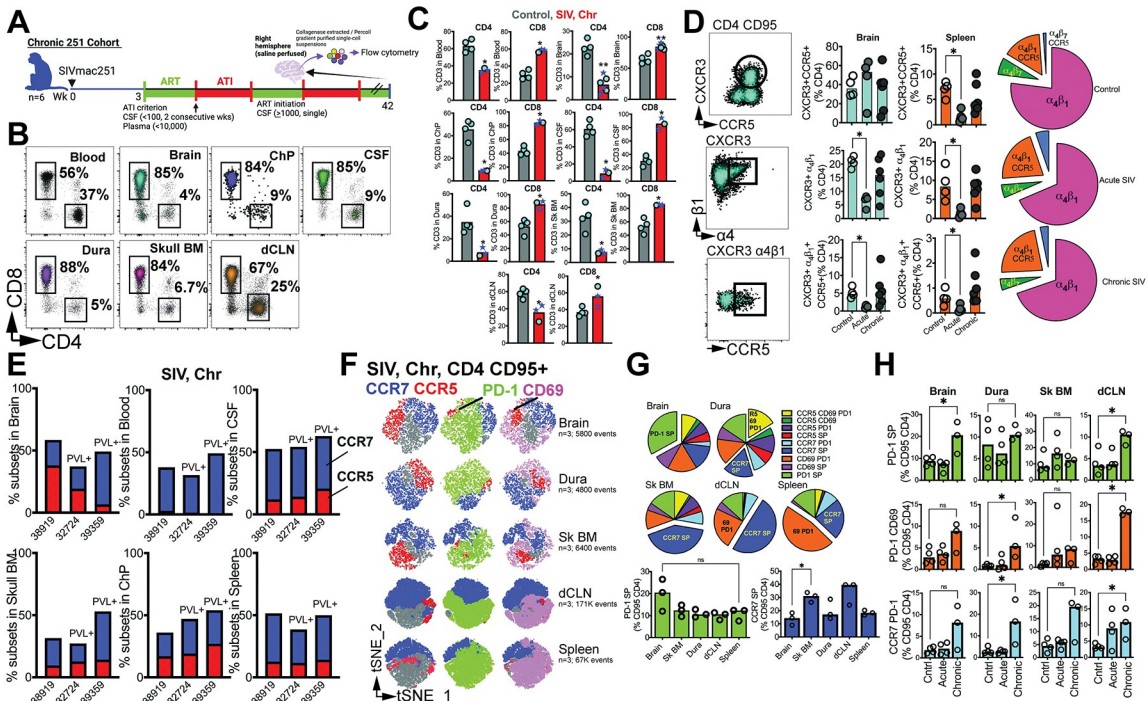

**Fig 8. Persistent CD4 depletion in CNS during chronic infection. (A)** Chronic 251 cohort assessed; flow cytometry analysis performed on 4/6 animals. **(B)** Flow cytometry plots illustrate frequencies of CD4 and CD8 T cells in blood, CNS tissues, and dCLN. **(C)** Bar graphs show T cell frequencies across blood, CNS tissues, and dCLN in control and Chronic SIV infected macaques (n = 4, plasma viral RNA+ shown in star symbols). **(D)** flow plots (top to bottom) show CD4 CD95 cells co-expressing CXCR3 and CCR5; $T_h1$ cells expressing $\alpha_4\beta_1$, and $\alpha_4\beta_1$ $T_h1$ cells expressing CCR5. Bar graphs show frequencies in brain and spleen and pie charts illustrate relative proportion of subsets in brain (n = 4). **(E)** Contingency plots show distribution of CCR5 / CCR7 CD4 T cells in chronic SIV infection (n = 3 based on criterion of CD4 events > 100 in all CNS tissues). **(F)** t-SNE plots gated on CD4+CD95+ cells expressing CCR7/CCR5/PD-1/CD69 (n = 3). **(G)** pie chart demonstrates proportion of CD4 T cells expressing combination of markers (CCR7/CCR5/PD-1/CD69). **(H)** shows distribution of specific subsets. Significant differences by one tailed Mann Whitney test, *, p< 0.05. Schematics were generated using BioRender.

were elevated in the dura and dCLN. Additionally, an increase in the CCR7+ PD-1+ CD4 T cell subset was noted in the dura and dCLN. Overall, the data show that there is an increase in CD4 T cells expressing PD-1 within the brain parenchyma and CNS during the chronic phase of infection, even with suppressive ART. This increase indicates ongoing antigen-mediated T cell stimulation in CNS, likely due to relatively short period of ART for 2–4 weeks prior to necropsy.

## Discussion

A deeper understanding of immune mechanisms driving viral establishment, persistence, and neuroinflammation holds the potential to improve the quality of life for those living with HIV by addressing neurological complications and cognitive impairments early and effectively. Our findings provide significant insights into the immune environment within the brain parenchyma during both acute and chronic HIV infection. Specifically, we have identified distinct subsets of activated CCR5+ CD4 T cells and resting CCR7+ CD4 T cells. This discovery, coupled with our existing knowledge of these cell subsets derived from lymphoid and mucosal tissues [48–51], strongly supports the hypothesis that the immune milieu in the brain facilitates both active viral replication and the persistence of viral reservoirs during R5-T cell tropic infections. Notably, our single-cell, phenotypic, and functional analyses demonstrate striking

parallels between CD4 T cells in the brain parenchyma and those within the spleen during acute SIV infection. Beyond the brain parenchyma, our research sheds light on the immune composition of brain border tissues, uncovering HIV target CCR5+ CD4 T cells in the choroid plexus, dura, and skull bone marrow. The presence of vRNA and vDNA in these areas, along with their interconnectedness, holds significant implications. These findings indicate that CSF viral loads can provide more comprehensive insights than previously appreciated. They may not only reflect viral activity in the brain parenchyma and blood but also indicate viral presence in the choroid plexus and skull bone marrow. Equally notable is the existence of a lymphoid niche composed of CCR7+ CD4 T cells expressing IL-7R in the CNS. This finding is important as homeostatic proliferation of these cells could contribute to the longevity of virally infected cells within the CNS. In summary, our work not only sheds light on the intricate CD4 T cell landscape within the CNS but also highlights the active responses of T cells in the brain to SIV infection.

In addition to CD4 T cells, our single-cell analysis of CD45+ cells from the brain demonstrates the presence of distinct CD8 $T_{CM}$ and $T_{EM}$ clusters during acute SIV infection. While we did not identify a definitive resident memory ($T_{RM}$) cluster [52] the expression of CD69 transcripts within $T_{EM}$ and CD69 surface expression strongly suggests the co-existence of effector and resident CD8 T cells within the rhesus brain parenchyma. Furthermore, CD69 alone, in the absence of CD103, is shown to be sufficient in identifying $T_{RM}$ in non-CNS tissues [53]. This indicates that T cells in the brain exhibit a spectrum of differentiation states and respond rapidly to SIV infection in the CNS. Notably, we observed the upregulation of the canonical pathogen-specific effector chemokine, CCL5, in the CD8 $T_{EM}$ cluster. Additionally, the presence of Ki-67, a marker for cellular proliferation, and increase in CD28-effector memory CD8 T cells in the brain is consistent with active recruitment of SIV-specific CD8 T cell effectors to the brain parenchyma during acute infection [54, 55]. Our single-cell analysis of the CD4 T cell cluster in the brain also uncovered crucial details into the immune response during acute SIV infection. We observed a significant downregulation of key genes, such as IL7R and CD4, indicating strong T cell activation. Additionally, the expression of important anti-inflammatory receptors, including ADORA2B (receptor for extracellular adenosine) and the glucocorticoid receptor NR3C1, was also reduced. Conversely, we also noted the activation of antiviral IFN genes, signifying the active engagement of CD4 T cells within the brain. Importantly, this antiviral gene expression pattern closely paralleled what was observed in CD4 clusters within the spleen, indicating a shared antigen-specific and bystander response pattern across infected tissues. Exploring clonal heterogeneity of T cell subsets will yield deeper insights into immune response dynamics within the CNS during infection.

Although we did not measure cell-associated vRNA in CD4 T cells in the CSF in our studies, we infer the contribution of productive CD4 infection to CSF viral burdens. This inference is supported by the fact that R5-T cell tropic viruses, such as SIVmac251, require high levels of CD4 to enter cells. Previous research has also shown rapid viral decay following ART initiation, which aligns with our observations, consistent with replication in short-lived T cell effectors [56] [57]. Recent studies have shown spliced cell-associated viral RNA in CSF CD4 T cells during the acute stages of HIV-1 infection and weeks 2, 4, and 8 in SHIV-infected macaques, supporting active viral transcription within CSF CD4 T cells [32]. Studies in PLWH during suppressive ART further support this model [32, 58]. Utilizing the T cell activation marker CD26 to distinguish HIV virions derived from CD4 T cells versus macrophages, Lustig et al. report T cell-derived virus in the CSF, even in individuals experiencing CSF escape [58]. Although compartmentalized virus in the CSF is attributed to viral replication in brain-resident myeloid cells [20], compartmentalized replication of R5-T cell tropic T/F virus in ART-naive PLWH has also been observed [57]. Altogether these data support the importance of

CD4 T cells in contributing to acute stage CNS viral burden and neuroinflammation, setting the stage for legacy effects.

The presence of HIV target CCR5+ CD4 $T_h$1 cells in immune-rich compartments within the CNS beyond the CSF highlights their potential role in supporting viral replication. Notably, higher levels of vRNA within the dura, compared to the parenchyma, might be attributed to transitory cells draining from the brain and subarachnoid space through dural lymphatic vessels, although the possibility of resident cells in this compartment cannot be ruled out [33, 59]. Another intriguing site is the choroid plexus stroma, positioned at the interface of peripheral blood and CSF, known to harbor macrophages and dendritic cells. The presence of CCR5 + CD4 T cells and vRNA in this tissue underscores its importance in terms of inflammation and viral evolution. Of particular significance is the potential establishment of reservoirs in the skull bone marrow niche of the CNS, which has access to the dura and brain, especially in the context of inflammation [60, 61]. Based on the mobilization of the myeloid niche from the skull bone marrow to the brain during inflammation [60], investigating whether a similar phenomenon occurs for T cells could yield valuable insights into their migration and role in seeding the brain with virally reactivated CD4 T cells. Furthermore, the homeostatic proliferation of CCR7+ CD4 T cells and clonal expansion within the marrow may contribute to reservoir maintenance, potentially enabling the transit of cell-free or cell-associated virus to border tissues of the brain and the CSF. Assessing viral sequences in sorted CCR5+ and CCR7+ subsets across CNS compartments and in peripheral subsets during acute and chronic infection under suppressive ART will provide deeper insights into their respective roles in promoting viral replication and persistence.

The strategic positioning of CXCR3+ CCR5+ CD4 T cells within the CNS—including the brain parenchyma, CSF, choroid plexus stroma, dura, and skull bone marrow—highlights their significance not only in neuroinflammation but also in sustaining CNS viral presence. In support of this concept, consistent elevation of CSF IP-10 levels compared to plasma throughout infection signifies a CNS environment primed for CXCR3-IP-10 mediated T cell ingress. This implies that the CNS remains receptive to CCR5+ CD4 T cell infiltration, potentially perpetuating viral presence. Equally pivotal is data showing that despite CD4 T cell depletion during infection, frequencies of CCR5 and CCR7 within the CNS remain relatively stable. This stability prompts consideration that underlying mechanisms sustain a pool of CCR5+ CD4 T cells within the CNS. The presence of CCR5+ CD4 T within the skull bone marrow, coupled with activation of the skull bone marrow niche during neuroinflammation, indicates possible conduit for cell-free or cell-associated viral migration from the skull bone marrow to the brain [61]. By investigating dynamics of viral sequences across different CNS compartments, future studies may uncover insights into CNS viral evolution and the role of CCR5+ CD4 T cells in driving this process.

Once inside the brain, T cell engagement with brain-resident innate immune cells can trigger immune activation. Indeed, the consideration of brain resident microglia and macrophages is paramount in understanding the full extent of viral dissemination and neuroinflammation within the CNS. As major immune cells expressing CD4 and CCR5, albeit at lower levels than CD4+ T cells, they become crucial targets for viral infection in the CNS [57, 62]. High responsiveness of microglia to IFNγ, a classic CD4+ $T_h$1 cell cytokine, can activate them, leading to the release of free radicals and inflammatory mediators like TNFα and IL-1β, ultimately contributing to neuronal death [63–65]. This ensuing inflammation, particularly the induction of IP-10, could then trigger waves of $T_h$1 CD4 T and CD8 T cell influx, further intensifying immune activation. Such a continuous cycle of viral replication and immune activation could contribute to viral persistence within the CNS. Conducting proof-of-principle studies to investigate these mechanisms and delineating the role of immune trafficking across

distinct CNS compartments is of import as they hold the potential to significantly advance our understanding of the cells involved in acute and chronic neuroinflammation, as well as viral persistence within the CNS.

While our findings significantly contribute to our understanding of CD4 T cell responses in the CNS following acute and chronic SIV infection, it is crucial to acknowledge limitations in our study. First, our primary focus on modeling sub-optimal ART adherence inherently restricts the applicability of our findings, particularly regarding CNS viral persistence and immune activation, to fully suppressed settings. Second, our study did not evaluate SIV-specific T cell responses or the differential distribution of SIV-specific CD8 T cells across various brain regions, encompassing white and gray matter, and border tissues or frontal and temporal regions of the brain. These unexplored aspects undoubtedly hold important implications for virological control in distinct regions of the brain parenchyma and CNS. Lastly, exploration of viral sequences in CSF compared to plasma temporally, and across distinct CNS compartments during acute and chronic infection is important. This is an aspect that has not been addressed but holds significant potential to yield crucial insights into the virological and immunological determinants of viral persistence and control within the CNS.

In conclusion, our findings provide insights into viral replication and immune responses within the CNS. They have important implications for understanding disease progression, viral persistence, and the challenges in eradicating the virus from the CNS.

## Materials and methods

### Ethics statement

All animals were bred and housed at the California National Primate Research Center (CNPRC) in accordance with the American Association for Accreditation of Laboratory Animal Care (AAALAC) guidelines. All studies were approved by the University of California, Davis Institutional Animal Care and Use Committee (IACUC).

### Rhesus macaques

For Acute 251 studies, four (1 male and 3 females, 11–17 years) colony-bred Indian origin rhesus macaques (*Macaca mulatta*) were utilized. For the Chronic 251 study, six adult (1 male and 5 females) Indian origin rhesus macaques (*Macaca mulatta*) were utilized. At study initiation, animals were 11.2–17.4 years of age with a median weight of 10.5 kg. Four rhesus macaques served as controls (12–16 years, 1 male and 3 females). All animals were SIV negative (SIV-), simian T-cell leukemia virus negative (STLV-), and simian retrovirus negative (SRV-); and had no history of dietary, pharmacological, or surgical manipulation (*S1 Table*).

### SIVmac251 infection

Rhesus macaques were infected intravenously with ($10^4$ TCID50) SIVmac251 (2017 stock from CNPRC at 2.5 x 10^4 $TCID_{50}$/mL, grown in rhesus peripheral blood mononuclear cells). Prior to inoculation, the virus was reconstituted in plain RPMI (virus stock: RPMI ratio 4:1) in total injection volume of 500 μl.

### Antiretroviral therapy

We formulated a triple-ART regimen described by Del Prete et al. [66] containing the nucleo (s/t)ide reverse transcriptase inhibitors emtricitabine (FTC) and tenofovir disoproxil fumarate (TDF) [from Gilead] with the integrase strand transfer inhibitor dolutegravir (DTG) [from GSK].

## Viral RNA quantification

Quantification of plasma, CSF, and tissue viral RNA and viral DNA were performed essentially as previously described [67] with assays performed in the Quantitative Molecular Diagnostics Core of the AIDS and Cancer Virus Program at Leidos Biomedical Research Inc., Frederick National Laboratory.

## ARV measurement

ARV concentrations in plasma, CSF, and tissue samples were quantified by LC/MS in the Clinical Pharmacology and Analytical Chemistry Core, UNC Center for AIDS Research as described previously [68].

## Specimen collection and processing

Cerebrospinal fluid, blood, and lymph node fine needle aspirates were sampled and processed as described previously [69, 70]. To isolate specific regions from the brain parenchyma, the saline-perfused brain was sectioned into 6mm coronal blocks, employing a fresh, sterile blade for each cut. Following the blocking procedure, clean forceps were employed to delicately extract the desired regions. For isolation of single cell suspensions from brain parenchyma, choroid plexus stroma, spinal cord, dura mater; tissues were mechanically dissociated and digested in DMEM with 0.25% trypsin and 5 units per mL of DNase I for 45 minutes at 37˚C. Digested tissues were homogenized using a pipette controller and 10mL serological pipette. The homogenized tissue was subsequently filtered through a metallic strainer followed by a 180μm nylon strainer and 100μm SMART strainer. Cells were washed in media and spun down at 1200 rpm for 10 minutes. Mononuclear cells were collected using a 21% and 75% Percoll gradient. Post gradient enrichment, cells were washed, counted, and up to $2x10^6$ million cells were stained with panel of fluorophore conjugated antibodies or cryopreserved for future analysis.

## Flow cytometry

Whole blood, CSF, and fine needle lymph node aspirates (FNA) were freshly stained and acquired on the same day following collection. Mononuclear cells obtained from necropsy tissues were either freshly stained and acquired the same day or stained following cryopreservation. For cryopreserved cells, samples were thawed at 37˚C and diluted in complete media. Cells were then washed and incubated in complete media with 2 units/mL of DNAse I for 15 minutes at 37˚C. Cells were washed with complete media and counted prior to staining. Whole blood samples were treated with BD FACS Lysing Solution (BD Bioscience) for 10 minutes and washed with 1X FACS buffer (phosphate buffered saline with 1.5mM sodium azide, 2% fetal bovine serum, 10mM EDTA) prior to surface staining. Antibodies for surface staining were prepared in Brilliant Stain Buffer Plus (BD Biosciences) and incubated with cells at 4˚C for 30 minutes and washed twice with FACS buffer. Sample acquisition and fluorescence measurements were performed on a BD Bioscience FACSymphony utilizing FACSDiva software (Version: 8. 0.1). Sample compensation, population gating, and analysis was performed using FlowJo (Version 10.8.1)

## Legendplex assay

This assay (BioLegend, USA) was conducted following the manufacturer's instructions to assess cytokine levels in plasma and CSF. Briefly, IP-10 (CXCL10, A6, cat# 740335), IL-8 (B7, cat#740344) and MCP-1(B9, cat#740345) multiplex beads were sonicated for two minutes in a

sonicator bath (Thermo Fisher, USA). These multiplex beads were then appropriately diluted in assay buffer and added to a V-bottomed plate. Plasma samples were diluted 2-fold in dilution buffer, while CSF samples were used without further dilution. Both sample types were added to the V-bottomed plate containing multiplex beads and left to incubate overnight at 4°C on a microplate shaker at 150rpm. The next day, the plate was washed twice with washing buffer and 25μl of detection antibody was added to each well followed by washing and incubation for 1 hour on a microplate shaker at room temperature (RT). Then, 25μl of SA-PE was added into each well directly and incubated for 30 min at RT. The plate was washed twice and resuspended in 200μL of wash buffer. The samples were acquired on a BD LSR Fortessa (BD Biosciences, USA) cell analyzer, with 900 events collected from each sample for analysis. The concentration (pg/mL) of IP-10 (CXCL10), IL-8, and MCP-1 was determined by extrapolating the values from the standard curve.

### Cerebrospinal fluid and serum biochemistries

Animal CSF and serum chemistries were quantified using a Piccolo Xpress Chemistry Analyzer (Abbott) with Piccolo BioChemistry Plus disks in accordance with manufacturer's instructions. Chemistry panel analytes included albumin, glucose, and total protein.

### Intracellular cytokine staining assay

The polyfunctionality of CD4 T cells was assessed using intracellular cytokine staining (ICS). Brain and spleen cells were stimulated with 1X Cell Stimulation Cocktail (PMA and ionomycin) (eBioscience, USA) along with R10 media in the presence of 0.2mg CD28/49d co-stimulatory antibodies (BD) per test. Unstimulated controls were treated with volume-controlled DMSO (Sigma-Aldrich). Cells were incubated in 5% $CO_2$ at 37°C and after 1 hour of stimulation, protein transport inhibitors 2ml/mL GolgiPlug (Brefeldin A) and 1.3ml/mL GolgiStop (Monensin) (BD, Biosciences, USA) was added to tubes and further incubated for 3 hours at 37°C, 5% $CO_2$. Following stimulation, cells were stained for ICS surface markers CD3, CD4, CD8, and CD95. Subsequently, the cells were fixed using cytofix/cytoperm for 10 min at 4°C, then permeabilized with 1X Perm wash buffer (BD, Biosciences, USA), and stained with intracellular markers TNFα, IFNγ, and IL-2 for 45 min. Finally, cells were washed and acquired the same day using a BD FACSymphony flow cytometer.

### Cell preparation for sequencing studies

Cryopreserved mononuclear cells from rhesus brain were thawed, placed in fresh complete media (For splenic cells: RPMI supplemented with 10% HI-FBS, 1% L-glutamine, 1% penicillin-streptomycin; For brain tissue derived cells: DMEM supplemented with 10% HI-FBS, 1% L-glutamine, 1% penicillin-streptomycin) and treated with 2 units/mL of DNase I (Roche Diagnostics) for 15 minutes at 37°C. Cells were washed in complete media and CD45+ cells isolated using CD45 magnetic bead separation for non-human primates (Miltenyi Biotec CD45 Microbeads non-human primate) in accordance with the manufacturer's protocol. Enriched CD45+ cells were stained for CD45 and a live dead marker for subsequent flow cytometric sorting. Live CD45+ cells were characterized and quantified on a BD FACSymphony cell analyzer and sorted utilizing a FACS Aria and suspended in RPMI for single cell RNA sequencing studies.

### Single cell RNA sequencing

Sample barcoding, assembly of gel-beads in emulsion (GEM), GEM reverse transcription, cDNA amplification and cleanup, and library construction were performed according to

the Chromium Next GEM single cell 3' v3.1 protocol from 10X Genomics. Sequencing was performed by SeqMatic LLC on a NovaSeq 6000 platform using S4 200 flow cells with paired end reads run in four replicates with an average of 111,000 reads per cell. Sample demultiplexing, generation of FASTQ files, sequence alignment, gene counting, and sample aggregation were performed using the Cellranger pipeline version 7.1.0. Samples that passed data quality control steps (removal of samples with low quality reads, low frequency of mapped reads, low number of reads per cell, high mtRNA signature), were used for subsequent analyses. Sequenced reads were aligned to the Mmul_10 genome reference for Rhesus macaque, and raw count matrices were generated which were used as the input to the Seurat integrated analysis pipeline (Seurat V4.3.0). Quality control was done at the gene and cell level accounting for the median number of genes, and mitochondrial gene percentage using quality control plots.

## Bioinformatics

To process the sequencing data, we performed sequence alignment to the reference genome of *M.mulatta* (Mmul_10) using the 10X Genomics protocol (CellRanger V.6.0). The generated cell-by-gene count matrix was used for downstream analysis using the Seurat based integrative analysis workflow. The filtered count matrix (percentage of mitochondrial reads <10, and gene expressed in more than 10 cells) was log-normalized, and the top variable genes were used to perform the graph-based cell clustering with a resolution of 0.5 and visualized using Uniform Manifold Approximation and Projection (UMAP). Cluster identity was assigned by a combination of approaches including identifying cluster-specific differentially expressed genes, expert knowledge, canonical list of marker genes, and automated annotations using immune reference atlas through SingleR. Differential gene expression (DEG) analysis of the different cell-types across conditions was performed using the functions from Seurat and were selected at a threshold of (adjusted P-value < 0.05, |log2 FC| > 0.25) based on Benjamini-Hochberg correction. Gene-set enrichment analysis, and functional annotation was implemented through clusterProfiler 4.0, and visualized using custom scripts. All downstream data analysis was performed using R v4.2.0. We used the monocle3 based workflow to estimate lineage differentiation between the cell population based on the experimental conditions. We extracted the subsets of the identified celltypes from our integrated Seurat object and further inferred the trajectory graphs. Using the defined root node ($T_{CM}$), we selected lineages based on the shortest path that connects the root node and the leaf node. After establishing the different lineages, we implemented a differential gene test to find genes that change as a function of pseudotime based on a combination of Moran's statistic and q-value and visualized using heatmaps and individual gene trajectory plots. To count the viral transcripts in the data, we built a custom reference using the CellRanger mkref pipeline. We downloaded the FASTA and created the GTF files of Simian immunodeficiency virus isolate SIVmac251.RD.tf5 (SIVmac251) and added it to the reference genome files of *M.mulatta* (Mmul_10). The customized reference was successfully created, and the generated count matrix which included the viral transcript was used in all steps of further downstream analysis. To calculate the percentage of SIV transcript expression in vRNA+ cells, we initiated the process by selecting cells that expressed the SIV transcript above a threshold of 1. Subsequently, we determined the proportion of all counts attributed to a subset of potential genes within these selected cells. The percentage for each cell was then computed as the count of the SIV transcript divided by the sum of all gene counts, multiplied by 100. All downstream data analysis was performed using R v4.2.0. Venn diagrams were created utilizing http://bioinformatics.psb.ugent.be/webtools/Venn/.

### In situ hybridization (ISH) and CD3 immuno-fluorescence

In situ hybridization (ISH) and CD3 immuno-fluorescence were carried out following a modified version of the manufacturer's protocol (Document Number 322452-USM and UM323100, ACD) for RNAscope ISH built on our previously established work [71] and the probe for SIVmac 239 (Catalog no. 405661) was custom-designed spanning Gag, Pol, and Nef genes to enhance sensitivity [72]. The procedure involved several steps using the RNAscope Multiplex Fluorescent Reagent Kit (ACD). Initially, four-micron deparaffinized paraffin sections underwent pretreatment with 1X Target Retrieval Buffer at 100˚C for 15 minutes, followed by RNAscope Protease Plus at 40˚C for 30 minutes before hybridization with probes at 40˚C for 2 hours. Subsequent signal amplification steps were conducted after hybridization. Detection of the signal was achieved using TSA Vivid fluorophore 5 (Cat# 323271, ACD) for 10 minutes at room temperature. For CD3 immuno-fluorescence, slides underwent an additional IHC staining process following RNAscope ISH. This involved overnight incubation at 4˚C with Rat polyclonal anti-CD3 (Abcam) at a 1:100 dilution. Detection of CD3 cells was facilitated by using Alexa Fluor 568 goat anti-rat IgG (Invitrogen). After DAPI staining, slides were cover-slipped with ProLong Gold anti-fade mounting agent (Invitrogen). In each ISH run, probe RNAscope Probe—SIVmac239 (Cat# 405661) was accompanied by probes for dihydrodipicolinate reductase (dapB) or RNAscope 3-plex negative control probe. Tissues from SIV-uninfected animals were also hybridized with the SIV probes to serve as negative controls. To ensure the quality and consistency of the ISH assay, RNAscope Probe—Mau-Ppib and RNAscope 3-plex positive control probe were employed as positive controls for RNA quality. Visualizations were carried out using appropriate filters, and images were captured with a Zeiss LSM800 confocal microscope and Zeiss Imager Z2 (Carl Zeiss).

### Statistical analyses

Wilcoxon signed rank test were used for paired analyses (i.e., longitudinal and within group comparisons). Mann-Whitney U-test were used for unpaired comparisons between animal cohorts/treatment groups. Tests were performed in GraphPad Prism (Version 9.5.1) with significance values denoted as follows: * $p < 0.05$, ** $p < 0.01$, *** $p < 0.001$, **** $p < 0.0001$. Schematics were generated using BioRender.

## Supporting information

**S1 Table. Non-human primate cohorts.**
(DOCX)

**S2 Table. Plasma and CSF vRNA in Chronic 251 cohort.**
(DOCX)

**S3 Table. Antibody reagents for flow cytometry analysis.**
(DOCX)

**S1 Fig. Gating strategy for CNS tissues in SIV unexposed controls.**
(TIF)

**S2 Fig. Gating strategy for CNS tissues in Acute SIV infection.**
(TIF)

**S3 Fig. CCR5/CCR7 dichotomy during homeostasis/Blood counts during acute SIV-mac251 infection. (A)** Bar graphs illustrate discrete distribution patterns of CCR5 and CCR7 on CD8+ CD95+ cells in blood, lymph node FNA, and CSF in Control cohort 2 (n = 12). **(B)**

Kinetics of body weight, white blood counts (WBC), red blood counts (RBC). **(C)** Kinetics of lymphocyte, CD4 T cells, monocyte, and neutrophil counts during first 4 weeks of SIVmac251 infection in Chronic 251 cohort (n = 6). Significant differences by Wilcoxon matched-pairs signed rank test, *, p< 0.05 **, 0< 0.01, ***, p< 0.01. For CD4 T cell counts, p value corresponds to 5/6 animals in gray.
(TIF)

**S4 Fig. Plasma and CSF cytokines during acute SIVmac251 infection.** Kinetics of IL-8 and MCP-1 measured by Legend plex assay during first 3 weeks of SIVmac251 infection in Chronic 251 cohort (n = 6). Significant differences by Mann Whitney test *, p< 0.05.
(TIFF)

**S5 Fig. CSF CCR5+ CD8 T cell frequencies do not decrease during acute SIV infection.** shows % CD8 T cells, % CD28- CD95+ CD8 T cells, %CCR5+ CD8 T cells, and % CD69 + CD8 T cells in CSF in Chronic 251 cohort (n = 6). Significant differences by one-tailed Wilcoxon matched-pairs signed rank test, *, p< 0.05.
(TIFF)

**S6 Fig. CSF parameters during acute SIVmac251 infection.** CSF albumin, protein, glucose, and glucose/albumin ratio during first 3 weeks of SIVmac251 infection in Chronic 251 cohort (n = 6).
(TIFF)

**S7 Fig. CCR5+ CD4 T cells populate parenchymal and border CNS tissues. (A)** Flow plots and **(B)** bar graphs show CCR5 expression on CD4+CD95+ T cells in controls, second bar graph shows CCR5 CD8 frequencies. Control 1 cohort (n = 4) assessed. **(C)** shows phenotype of CCR5+ CCR7- versus CCR7+CCR5- cells. **(D)** bar graph of CCR5 /CCR7+ CD4 T cell subset frequencies in brain, dura, and skull bone marrow at 3 weeks post SIV. Acute 251 (n = 4) cohort assessed.
(TIF)

**S8 Fig. SIVRNA localizes with immune clusters in brain and spleen.** (**A**) Genes enriched in $T_{EM}$ and $T_{CM}$ clusters in control and SIV brain (p.adj < 0.05). (**B**) UMAP of immune clusters in brain and vRNA expression in clusters in control and SIV. (**C**) UMAP of immune clusters in spleen and vRNA expression in clusters in control and SIV. Acute 251 (n = 4) cohort assessed.
(TIF)

**S9 Fig. Volcano plots of DEG in SIV brain.** Volcano plots of immune clusters show genes up and downregulated in SIV relative to controls. Genes meeting padj and fold-change cut-off are denoted in red. Acute 251 (n = 4) cohort assessed.
(TIF)

**S10 Fig. Pseudotime plots of Control and SIV brain.** (**A**) UMAP. (**B**) pseudotime trajectory comprising of distinct immune clusters shows Lineages 1–3. Lineage 4 comprised only of CD8 $T_{EM}$. (**C**) shows heat map comprising of T cell clusters from Lineage 1. Acute 251 (n = 4) cohort assessed.
(TIF)

**S11 Fig. Kinetics of plasma and CSF vRNA, plasma IP-10 and CSF IP-10.** Green bar indicates duration of ART with FTC, TDF, and DTG. FTC = emtricitabine, TDF = tenofovir disoproxil fumarate (TDF), DTG = dolutegravir. Chronic 251(n = 6) cohort assessed.
(TIFF)

## Acknowledgments

The authors extend their sincere appreciation to several individuals whose invaluable contributions have been pivotal in the successful execution of the studies. Our thanks to Wilhelm Von Morgenland and Miles Christensen, as well as the dedicated CNPRC SAIDS team, for their exceptional coordination of macaque studies, diligent animal care, and animal support. We also express our gratitude to the CNPRC Veterinary Staff for animal care and assistance. We thank GILEAD and GSK for their generous provision of ART drugs. Special thanks to Jennifer Watanabe, and the members of Koen Von Rompay's laboratory for their invaluable technical support during necropsies. We would like to convey our appreciation to Andradi Villabos for assistance provided during necropsies and for the acquisition of flow cytometry data. The authors acknowledge the NIH Tetramer Facility for providing the Gag CM9 reagent for the studies.

## Author Contributions

**Conceptualization:** John H. Morrison, Smita S. Iyer.

**Data curation:** Sonny R. Elizaldi, Anil Verma, Dhivyaa Rajasundaram, Chase E. Hawes, Yashavanth Shaan Lakshmanappa, Smita S. Iyer.

**Formal analysis:** Sonny R. Elizaldi, Anil Verma, Dhivyaa Rajasundaram, Chase E. Hawes, Yashavanth Shaan Lakshmanappa, Jeffrey D. Lifson, Smita S. Iyer.

**Funding acquisition:** John H. Morrison, Smita S. Iyer.

**Investigation:** Sonny R. Elizaldi, Anil Verma, Zhong-Min Ma, Sean Ott, Dhivyaa Rajasundaram, Chase E. Hawes, Yashavanth Shaan Lakshmanappa, Mackenzie L. Cottrell, Zandrea Ambrose, Smita S. Iyer.

**Methodology:** Sonny R. Elizaldi, Anil Verma, Zhong-Min Ma, Sean Ott, Dhivyaa Rajasundaram, Chase E. Hawes, Yashavanth Shaan Lakshmanappa, Mackenzie L. Cottrell, Jeffrey D. Lifson, Smita S. Iyer.

**Project administration:** John H. Morrison, Smita S. Iyer.

**Resources:** Angela D. M. Kashuba, Jeffrey D. Lifson, John H. Morrison, Smita S. Iyer.

**Supervision:** Angela D. M. Kashuba, Jeffrey D. Lifson, John H. Morrison, Smita S. Iyer.

**Writing – original draft:** Smita S. Iyer.

**Writing – review & editing:** Sonny R. Elizaldi, Anil Verma, Zandrea Ambrose, Jeffrey D. Lifson, Smita S. Iyer.

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
