## [Decision Letter · Decision Letter 0]

29 Sep 2023

Dear Dr. Iyer,

Thank you very much for submitting your manuscript "CD4 T cell responses in the rhesus CNS during SIV infection" for consideration at PLOS Pathogens. As with all papers reviewed by the journal, your manuscript was reviewed by members of the editorial board and by several independent reviewers. In light of the reviews (below this email), we would like to invite the resubmission of a significantly-revised version that takes into account the reviewers' comments.

All reviewers found merit in the manuscript and recognized this was a relatively unexplored, but potentially very important, biological site. However, all reviewers had suggestions that should be addressed.

We cannot make any decision about publication until we have seen the revised manuscript and your response to the reviewers' comments. Your revised manuscript is also likely to be sent to reviewers for further evaluation.

Sincerely,

Jason M. Brenchley

Academic Editor

PLOS Pathogens

Susan Ross

Section Editor

PLOS Pathogens

Kasturi Haldar

Editor-in-Chief

PLOS Pathogens

orcid.org/0000-0001-5065-158X

Michael Malim

Editor-in-Chief

PLOS Pathogens

orcid.org/0000-0002-7699-2064

All reviewers found merit in the manuscript and recognized this was a relatively unexplored, but potentially very important, biological site. However, all reviewers had suggestions that should be addressed.

Reviewer's Responses to Questions

**Part I - Summary**

Reviewer #1: The study by Elizaldi et al seeks to map immune and viral dynamics in fluid and tissue-associated compartments of the CNS in healthy, acutely and chronically infected macaques infected with SIVmac251. CNS tissues likely are a non-negligible component of the reservoir and a site of immune abnormalities associated with cognitive impairments. It thus warrants studying this anatomical site in the SIV macaque model. Overall the study shows that regions of the CNS, particularly those of the brain and border tissues, are target-cell rich and active sites of SIV replication and inflammation. While this has already been described to some extent the use of comprehensive cellular, single cell transcriptome, and reservoir assessments across multiple CNS tissues provide a level of detail that I believe advances on these previous studies, albiet if it mostly descriptive in nature. Part of this study set out to model sub-optimal adherence to ART with 2-3 ATIs and animals were thus viremic for the majority of the time on treatment. My main concern is that it is unclear how these findings may relate to reservoir or immune abnormalties that persist in fully-suppressed settings which is the more clinically pertinent question. The study nevertheless characterizes the CNS in viremic settings with rigor and could be suitable for plos pathog. I have a major comment on SIV-specific CTLS and several minor comments.

Reviewer #2: The manuscript by Elizaldi et al provides a very detailed and extensive characterization of T cells in different compartments of the CNS in uninfected and SIV infected rhesus macaques as well as the distribution of viral RNA and DNA. Studies of NeuroHIV in people are limited to analyses of CSF and autopsy samples. Therefore, studies in animal models of HIV/SIV infection in the CNS to understand T cell and virus dynamics are of importance. This study analyzed acutely infected and chronically infected animals. The chronically infected group of animals were followed for 42 weeks post-exposure and were treated with ART at three weeks post-exposure. Interestingly, the animals also underwent 1-3 rounds of antiretroviral therapy interruption during the course of the experiment to mimic intermittent or poor adherence to ART. While the animals were treated with ART for 2-4 weeks prior to necropsy, it is important to note that some of the findings in this group like incomplete recovery of CD4 T cells in the brain and high levels PD1 expression may be due to multiple rounds of ATI in the animals resulting in virus replication and antigen production. The results obtained here may not reflect what occurs in the CNS of people living with HIV that are durably suppressed with ART.

Reviewer #3: In the manuscript "CD4 T Cell Responses in the Rhesus CNS during SIV Infection", by Elizaldi et al., the authors characterized the immunological and viral dynamics in the central nervous system (CNS) and border tissues during acute SIV infection, SIV infection under ART, and ART interruption using two cohorts of CCR5-tropic SIVmac251-infected rhesus macaques (RMs) and a control cohort of n=4, age-matched SIV-unexposed RMs. Specifically, the evaluated n=4 RMs during the acute phase of infection (week 3 p.i.) and n=6 RMs in the chronic phase (followed for up to 40 weeks p.i.). The chronic cohort underwent multiple cycles of ART and ATI; ART was re-initiated for 2 to 4 weeks before necropsy.

Among their key findings, a higher frequency of CCC5+CCR7- CD4+ T cells and CD8+ T cells was observed in the CSF compared to blood and FNA in SIV-unexposed RMs. CXCR3+ and CCR5+ CD4+ T cells were more abundant in brain parenchyma and skull bone marrow compared to the spleen. Upon SIV infection, there was a decrease of both total and CCR5+ CD4+ T cells in the CSF. After initiating ART, total and CCR5+ CD4+ T cells were partially restored. Furthermore, most PD-1 and CD69 expression was detected on CCR5+ CD4+ T cells, and the co-expression of these three surface markers was more prominent in the brain and dura mater. The authors also conducted single-cell RNA-seq on sorted CD45+ cells from four SIV-infected RMs and detected SIV transcripts not only in CD4+ T cells but also in clusters of monocytes/macrophages in the brain. The choroid plexus and dura mater harbored the highest SIV DNA and RNA levels, whereas the vRNA/DNA ratio was highest in white matter regions. The authors demonstrated that RMs exhibited a similar viral decay rate in plasma and CSF after starting ART, and that plasma and CSF vRNA rebounded with similar kinetics in most animals after discontinuing ART.

The strength of the article is the extensive phenotypic and transcriptomic characterization of CD4+ T cells from the brain and surrounding tissues during SIV infection, which make this work novel. It is noteworthy that these tissues were also compared with multiple peripheral tissues. I do have some comments the authors should address to improve the clarify and significance of the manuscript.

Reviewer #4: In this study, the authors are taking a deep look at the dynamic, location, function, susceptibility to infection and phenotype of the CD4+ cell population in the brain and adjacent tissues while comparing with more distal sites and blood.

The authors have used numerous cutting edge assays in a well designed animal study and have made important reports as the identification of distinct CD4+ subsets: CCR5+ and CCR7+, suggesting that the CNS can be both the site of active viral replication and latency. It was also reported the presence of viral genome in the brain parenchyma and adjacent tissues like skull BM and Dura matter identifying those tissues as potential reservoirs. The single cell analysis of the CD4+T cells from the brain identified key genes showing clear activation. Finally the parallelism between T cells isolated from the brain and T cells isolated from spleen is showing that those cells have a similar biological response to SIV.

This manuscript contains lot of data and can benefit of some trimming to provide a clearer message. I hope that some minor suggestions can improve this manuscript.

**Part II – Major Issues: Key Experiments Required for Acceptance**

Reviewer #1: - THe authors mention SIV-specifc responses as data not shown but the authors should really include this and further provide a rigorous characterization or any relation to CNS reservoir. Particularly because (1) the authors show high prevalence of CCR5+ CD8 T cells as they do for CD4s, indicating potential shared trafficking mechanism, (2) transcriptionally-active vDNA and thus likely high level of antigen, and (3) CD8 T cell infiltration in multiple CNS tissues in the chronic stages. This has implications for how actively particular CNS tissues are surveilled by viral-specific CTLs, the lack thereof may indicate some degree of sanctuary for SIV persistence.

Reviewer #2: • For the data presented in Figures 1 and 2 and the corresponding supplementary figures, it is not always clear in the manuscript text, figure legends, and figures if the two infection groups (acute and chronic) are being analyzed together or if data from only one group is shown. This should be clear to the reader both in the manuscript text and in the figure legends. In addition, the authors should include the group numbers (n) in each figure legend (all figures). For the flow cytometry analysis in Figure 1C and Figure S1 there are ~12 data points. However, there are only 4 animals in the acute group and 6 animals in the chronic group. Are there multiple data points being shown for some of the animals? If so, this should be indicated in the figure legend.

• For the single cell transcriptomic analysis, the authors state that the CD8 TCM cluster expressed transcription factors ID2 and JUN and the CD8 TEM cluster expressed the transcription factors ZEB2 and STAT4 and CD69 and Ki67. Based on the presentation of the data in Figure 4C, there does not seem to be much difference in ID2, JUN, or STAT4 expression between CD8 TCM and CD8 TEM cells, especially in the analysis of cells isolated from SIV infected animals. There is also little difference in CD69 expression, if anything it would appear that there is slightly more CD69 expression in CD8 TCM in SIV infected animals.

• Figure 5B,C characterizes the genes with upregulated expression in SIV infected animals. Were there any notable genes or pathways with downregulated expression in SIV infected animals? There is mention of downregulated genes in the discussion but not in the results.

• In Figure 7A and in the figure legend, please indicate which colored line represents the CSF viral load and which represents the plasma viral load. It would appear that blue=CSF and red=plasma. However, if that is the case then it would appear that the first ATI event was initiated in RM 37274 and RM 38359 before the criteria of two consecutive CSF viral loads below 100 copies/ml.

• The manuscript text indicates that RM 37274 was Mamu-A01 positive and had notable CSF viral suppression even without ART but Figure 7a indicates that RM 38889 and RM 36511 were Mamu-A01 positive and it would appear that RM8889 maintained suppression in CSF without ART. Were the labels for RM 37274 and RM 8889 switched? The authors speculate that there was CD8 T cell mediated control in the CSF. Was the virus in the CSF and plasma sequenced to see if there was compartmentalization and/or the emergence of any mutations that would make the virus in CSF less fit?

• In humanized mice, HIV infection resulted in lower CD4 T cell counts and higher CD8 T cell counts in the brain (PMID: 29863499). Following ART treatment, the CD4 T cell counts in the brain returned to levels similar to uninfected controls. In this study, restoration of CD4 T cell levels was not observed in ART treated macaques. This difference could be due to the fact that animals in this study underwent multiple rounds of ATI and were only on ART for 2-4 weeks prior to necropsy and/or because CD4 T cell levels are reported as a percentage of T cells which is influenced by the actual numbers of CD4+ and CD8+ T cells.

• Figure 8 shows that significantly more CD4 T cells express PD1 in the brain or dura (either PD1+ or PD1+CD69+) indicating that there is ongoing antigen stimulation in the CSF even with suppressive ART. It is important to note that the animals had multiple rounds of ATI and were only on ART for 2-4 weeks prior to necropsy. This is a different scenario compared to a person who is on durable suppressive ART for months to years.

Reviewer #3: 1. The authors evaluated the functionality of CD4+ T cells after PMA stimulation in the uninfected cohort (Figure 2H). It would be informative if the authors also evaluated the functionality of CD4+ T cells in the acute and chronic cohorts.

2. The authors made multiple statements that there is persistent CD4 depletion in brain parenchyma during chronic infection. It is important to clarify that their study included short ART and multiple cycles of ART interruption, thus it is not designed to determine CD4 reconstitution under prolonged ART. Indeed, in Figure 1E there is a partial reconstitution of total and CCR5+ CD4+ T cells after only 4 weeks of ART. Also, findings related to the viral reservoir presented in Figure 7E and F needs also to be discussed in this context (for example, the sentence . . . underscoring the challenges in eradicating HIV from the CNS, even with effective ART). This is not a model of effective ART.

3. 443-447: The authors indicate that there was a rapid decay of CSF viremia on ART up to week 6, suggesting viral output primary due to short lived cells. Then, they added that a second phase of decay was observed in 3 animals, indicating the involvement of long-lived cells as macrophages. First, there is no evidence this is related to macrophage, it can well be longer lived CD4 memory cells. Second, while there are animals in which VL slightly increased on ART, it is difficult to identify 3 animals in which the decay was slower as compared to week 6. Furthermore, all animals but one (week 10) have been on ART only until week 7 or 8. I don’t think the study design and the data are supporting that conclusion.

4. Fig 2H: the authors described the increased levels of IFNγ single-positive cells in the brain, as compared to spleen, but did not mention the lower levels of IL-2+ and TNFa/IL-2 double positive cells, that is the most robust difference between brain and spleen. This needs to be described, and the authors should consider how those differences impact on their sentence “unequivocally demonstrating that Th1 cells in the brain are highly functional (Figure 2H).

5. For Figure 4, the authors used RNAscope to confirm the PCR data, revealing vRNA with CD3+ cells in the CNS parenchyma. Viral transcripts were found also in monocyte/macrophage clusters, although at lower level. Does the RNAscope analyses confirm that data as well?

6. Based on the flow cytometry data, it will be interesting if the authors can use their single-cell data to examine transcriptional differences among CCR5+ and CCR7+ CD4+ T cells in the brain of both the unexposed and acute infection cohorts.

Reviewer #4: I would like to raise a few concerns/questions that I hope can improve this really reach and detailed manuscript.

- It would have been valuable to add the set of data regarding the Gag-specific CD8T cells mentioned on page 9.

- In general, across the manuscript the authors are using "brain parenchyma" to refer to brain tissue. It would be more accurate and informative to actually have the author naming the area of the brain used for each analysis.

- With the comment made by the authors on page 10 regarding the importance of T cells in infection but also of free virion, The absence of cell associated VL in the CSF samples needs to be raised. would you have enough cells in the CSF to run the assay?

- The authors are referring to the immune cell population in the brain as a "robust immune environment rich in T cells". Unless I missed it out of the many figures and data, there is no absolute count of the leukocytes in the brain done on the naive or infected animals? there is % of each population gated out of CD3+ cells but not an actual absolute count of each immune cell population before and after infection. Without those data it is difficult to judge of the robustness of the immune cell population in such a large tissue. The

A few interesting points were not discussed like:

- The abundance of CD8T cells in the brain compare to CD4 in SIV negative animals.

- there is no B cells found in the brain of the acute infected RM. Numerous studies have reported infiltration of B cells during inflammation/infection- Did the authors look for B cells?

- Can the authors give their opinion on the absence of DC in the SIV naive animals.

**Part III – Minor Issues: Editorial and Data Presentation Modifications**

Reviewer #1: - The authors may have mislabeled the summary data in Figure 1C. Representative dot plots show 46% CCR5 but the summary data for CSF has a median in the single digits.

- If the authors are suggesting rapid influx into the CNS during acute SIV then it would be relavent for the authors to show data on the levels of the R5 ligand MIP-1b in the CSF.

- CCR5+ cells appear depleted in CNS tissue, do the target cell densities at these various tissues relate in any way to the CNS or plasma VL?

- Figure 4D. For mapping the SIV transcripts to single cells it would be helpful to provide info on the number and percentage of cells from each cluster that were positive for SIV RNA.

- Figure 4D, E. For validation of viral read transcripts It is unclear if a particular threshold was set for read positivity to call a cell positive for SIV RNA, and whether identical thresholds were used for the representative control and SIV sample as the scales appear to be different. As internal controls, the authors should overlay viral read mappings against the entire annotated CD45+ population of an SIV animal as in 4B and not just the Monos/CD4s to show a lack of viral transcripts in refractory populations such as NK or CD8 T cells.

- Figure 5A the gene annotations do not align entirely with their corresponding rows and towards the middle of the heatmap particularly it is difficult to determine which gene each row is representative of.

- Figure 7A needs a legend indicating the lines corresponding to plasma or CSF viremia.

- While ART was effective at suppressing viremia in the CNS, some of these animals at necropsy still had plasma viremia. This makes for a somewhat heterogenous cohort and it would be informative to denote the data points that correspond to viremic animals in Figure 8 when describing immune abnormalities at the chronic timepoint.

Reviewer #2: • Figure 1C, it would appear that the labels for FNA and CSF need to be switched in the graph showing %CCR5+CCR7-.

• The manuscript text states that CD4 T cell counts in the blood were decreased at week 4. However, there is no graph in Figure S1 or Figure 1 that shows CD4 T cell counts in the blood. Figure 1J shows the CD4/CD8 T cell ratio.

• The legend for Figure S1 states that the kinetics of lymphocyte, monocyte, and neutrophil counts is shown for the first 3 weeks of infection. However, the last time point in the graphs is week 4.

• It would be helpful to the reader to have a more detailed explanation of what is being shown in Figure 5E in the legend.

• The upper and middle right flow plots on Figure 8a are not labeled with the sample that they represent.

Reviewer #3: 1. The authors need to work on the flow of the manuscript as it lacks some cohesion. For example, in figure 1 the authors went from describing phenotypically unexposed RMs to describe the kinetics of viral loads and inflammation during acute infection and then to describe the effect of ART. Then in figure 2 they come back to describe the phenotype of unexposed RMs.

2. Fig 1C is inverted, the graph is showing higher levels of CCR5+CCR7- in FNA than CSF

3. Line 262: the majority of CXCR3+ cells in the CNS were α4β1+. According to the representative staining, a minority of the CXCR3+ in the CNS are a4b1+, 22%, with large majority of cells being B1+ but a4-. Please clarify. Similarly, line 310: The brain predominantly had CCR5 CD4 T cells; only 12% are reported as CCR5 (single) positive in the representative staining of Fig 3B. The authors should choice an animal more representative.

4. 440: Remarkably, 37274 MauA01+ demonstrated notable CSF viral suppression even without ART …. The animal with CSF viral suppression is indicated as 38889 in Fig 7. Please correct accordingly.

5. The authors should include supplemental figures or files with the complete gating strategies used to generate representative dot plots presented in the figures throughout the manuscript for each tissue.

Reviewer #4: - In my opinion the tittle is a little restrictive- the authors are looking at much more than the CD4T cell responses, like localization, phenotype, sensitivity to infection... this is more of a deep analysis of the CD4+T cells in the CNS.

- Figure 1: general observation, the large dots and lines makes the reading of the graph really hard and I could not tell how many animals were plotted on all those graphs.

(1A) route and dose on graph can be valuable information.

(1C) gating strategy can be removed to make panel more clear, the bar graph below it is sufficient.

(1I) is not essential plus the green color for the CSF doesn't match with the blue color used for all other figure in this panel.

(1K) can the authors add the same graph for CD8T cells.

Is there any correlation between cell population loss and blood and CSF VL?

- Figure1S: maybe the weight graph can be replace by CD4 count. Why there is not 10 animals? X axes has 4 weeks as end point was it not 3 weeks?

- entire paragraph on page 12 related to CD28 CD95 cells is in my opinion not relevant in this manuscript that has a lot of other data.

- Can the authors explain how the Choroid plexus was collected for VL? the number of vRNA and vDNA copies seam extremely high for a compartment of the brain that is only and strictly composed of epithelial cells secreting CSF. Was the Choroid plexus pulled out from the hemisphere at different location or was it cut off with surrounding parenchyma which can explain the results obtained by qPCR.

- Legend on graphs would be easier to read if on the side and colored boxes instead of ID and designation with a line.

-The ISH method needs explanation.

First the probe used for RNAscope as no information on the ACD website of the part of the genome targeted- the authors mentioned using a GAG probe in the figure legend but their is no other information (sequence targeted and number of Z pairs) in the method or on ACD probe catalogue. Plus the ref cited is a short communication with no M&M. Why would the authors not use a probe covering a larger part of the genome, especially when looking for rare events in a really large tissue?

The legend of the figure says "shows SIV RNA in parenchyma and perivascular regions of the brain" not sure where we are as there is no other specific marker or phase contrast in this picture.

It would have been more significant to have a picture of CD4 and myeloid cells combined to the RNAscope and ideally DNAscope.

PLOS authors have the option to publish the peer review history of their article (what does this mean?). If published, this will include your full peer review and any attached files.

Reviewer #1: No

Reviewer #2: No

Reviewer #3: No

Reviewer #4: No
---

## [Decision Letter · Decision Letter 1]

13 Nov 2023

Dear Dr. Iyer,

Thank you very much for submitting your manuscript "Deep Analysis of CD4 T cells in the Rhesus CNS during SIV infection" for consideration at PLOS Pathogens. As with all papers reviewed by the journal, your manuscript was reviewed by members of the editorial board and by several independent reviewers. The reviewers appreciated the attention to an important topic. Based on the reviews, we are likely to accept this manuscript for publication, providing that you modify the manuscript according to the review recommendations.

One reviewer had a few minor issues which should be addressed.

Sincerely,

Jason M. Brenchley

Academic Editor

PLOS Pathogens

Susan Ross

Section Editor

PLOS Pathogens

Kasturi Haldar

Editor-in-Chief

PLOS Pathogens

orcid.org/0000-0001-5065-158X

Michael Malim

Editor-in-Chief

PLOS Pathogens

orcid.org/0000-0002-7699-2064

One reviewer had a few minor issues which should be addressed.

Reviewer Comments (if any, and for reference):

Reviewer's Responses to Questions

**Part I - Summary**

Reviewer #1: My comments are adequately addressed and I have no others. Although the model may bear some resemblance to full virologic suppression, emphasizing the sub-optimal adherence nature of this study is more accurate than relating these findings to long-term persistence and it is importantt that the authors addressed this. They should be commended for their work.

Reviewer #2: This manuscript represents a very comprehensive analysis of the T cell compartment in the brain of uninfected and SIV-infected Rhesus Macaques. I thank the authors for addressing the points raised previously and making the requested modifications to the manuscript and figures. I do have additional points that I think should be addressed to improve the clarity of the manuscript.

Reviewer #3: The authors satisfactory addressed my previous comments.

Reviewer #4: The authors are bringing the light on the immune cells present in the CNS with focusing on the characterization of the CD4 + T cells. By using the NHP model and cutting edge assays, the authors are reporting an extensive phenotypic and transcriptomic of the CD4+ T Cells which is important information for the field.

The authors edited their first version of the manuscript in a relevant way by answering reviewers comments, concerns and by stating the limitation of their study. This manuscript will be really informative for all people studying SIV/HIV infection in the CNS.

**Part II – Major Issues: Key Experiments Required for Acceptance**

Reviewer #1: (No Response)

Reviewer #2: (No Response)

Reviewer #3: (No Response)

Reviewer #4: No major issue

**Part III – Minor Issues: Editorial and Data Presentation Modifications**

Reviewer #1: (No Response)

Reviewer #2: Introduction, page 6, line 138. Please revise to indicate that CD4 T cell depletion in the CNS was not rescued by suboptimal ART.

Results, page 7, line 163. I believe the text should state that the chronic cohort was followed for 42 weeks not 40 weeks.

Results, page 12, lines 295-297. Text indicates that the patterns of CCR7 and CCR5 expression on CD4+CD95+ T cells was similar between the CSF and other CNS tissues with the levels of CCR7 being lower in the brain parenchyma. However, in the dura the levels of CCR7 are also lower with more CCR5+ cells compared to CCR7+ cells with the exception of one animal.

Results, page 15, lines 376-378. The text states that the majority of the vRNA-positive cells were concentrated within the CD4 and monocyte clusters. I agree that the majority of the vRNA-positive cells clustered with CD4 T cells but in both the brain and spleen it would appear that there was more signal from the CD8+ T cell clusters (which are not infected with HIV) than the monocyte cluster in Figure S8 B-C. Could the signal in the myeloid and CD8 T cell clusters be background signal?

Results, page 15 lines 379-384. The text states that in the brain 11 vRNA+ cells were associated with CD4 T cell clusters and 1 vRNA+ cell was associated with monocyte/macrophage clusters and that in the spleen 4 vRNA+ cells were present in the CD4 T cell and monocyte clusters. However, Figure 4D-E indicate that in the brain 14 vRNA+ cells were associated with CD4 T cell clusters and 4 vRNA+ cell was associated with monocyte/macrophage clusters and in the spleen 8 vRNA+ cells were associated with CD4+ T cell and 3 vRNA+ cells were associated with monocytes/macrophages.

In Supplementary Figure 10, three lineages are shown in panel B but the results indicate that there were four distinct lineages identified by the pseudotime approach. Also, what are the two lineages that represent the transition from TCM to TEM in Supplementary Fig. 10B?

In regard to IP-10 CSF levels as a biomarker of ongoing viral replication (Results, page 21, lines 533-536), in Supplementary Figure 11, the levels of IP-10 in CSF do vary a little overtime but there doesn’t appear to be a consistent pattern where they go down or up during ART treatment.

Results, Page 21 lines 539-541. Please make it clear that the chronic experiment reflects the effect of suboptimal ART on CD4+ T cell depletion.

Much of the text in Figure 5E is too small to read. There are some genes that are shown in a larger font but there are other genes mentioned in the manuscript text that are too small to read in the figure.

The text in Supplementary Fig 10B is too small to read.

Results, page 18 line 454. Should the median vRNA copies in the ChP read 5x10^5 and not 0.5x10^5?

Please carefully check the references to make sure that they are correct. For example, the new text on page 22, lines 546-551 that discusses the results obtained in humanized mice references a paper on HIV infection of tissue resident memory CD4+ T cells.

For the description of Figure 8 in the results section, line 542 should reference Figure 8 A-B, line 544 should reference Figure 8C, line 558 should reference Figure 8D, line 560 should reference Figure 8E, line 562 should reference Figure 8F-G and line 564 should reference Figure 8H.

Reviewer #3: (No Response)

Reviewer #4: No minor issue

PLOS authors have the option to publish the peer review history of their article (what does this mean?). If published, this will include your full peer review and any attached files.

Reviewer #1: No

Reviewer #2: No

Reviewer #3: No

Reviewer #4: **Yes: **Claire Deleage

Figure Files:

Data Requirements:

Reproducibility:

References:

---

## [Editor Report · Decision Letter 2]

20 Nov 2023

Dear Dr. Iyer,

We are pleased to inform you that your manuscript 'Deep Analysis of CD4 T cells in the Rhesus CNS during SIV infection' has been provisionally accepted for publication in PLOS Pathogens.

Best regards,

Jason M. Brenchley

Academic Editor

PLOS Pathogens

Susan Ross

Section Editor

PLOS Pathogens

Kasturi Haldar

Editor-in-Chief

PLOS Pathogens

orcid.org/0000-0001-5065-158X

Michael Malim

Editor-in-Chief

PLOS Pathogens

orcid.org/0000-0002-7699-2064
---

## [Editor Report · Acceptance letter]

29 Nov 2023

Dear Dr. Iyer,

We are delighted to inform you that your manuscript, "Deep Analysis of CD4 T cells in the Rhesus CNS during SIV infection," has been formally accepted for publication in PLOS Pathogens.

Best regards,

Kasturi Haldar

Editor-in-Chief

PLOS Pathogens

orcid.org/0000-0001-5065-158X

Michael Malim

Editor-in-Chief

PLOS Pathogens

orcid.org/0000-0002-7699-2064